# Intranational synergies and trade-offs reveal common and differentiated priorities of sustainable development goals in China

Qiang Xing [1,2], Chaoyang Wu [3,4] ✉, Fang Chen [1,2,4] ✉, Jianguo Liu [5], Prajal Pradhan [6,7], Brett A. Bryan [8], Thomas Schaubroeck[9], L. Roman Carrasco [10], Alemu Gonsamo [11], Yunkai Li[12], Xiuzhi Chen [12], Xiangzheng Deng [3,4], Andrea Albanese [13], Yingjie Li [5,14] & Zhenci Xu [15]

Accelerating efforts for the Sustainable Development Goals requires understanding their synergies and trade-offs at the national and sub-national levels, which will help identify the key hurdles and opportunities to prioritize them in an indivisible manner for a country. Here, we present the importance of the 17 goals through synergy and trade-off networks. Our results reveal that 19 provinces show the highest trade-offs in SDG13 (Combating Climate Change) or SDG5 (Gender Equality) consistent with the national level, with other 12 provinces varying. 24 provinces show the highest synergies in SDG1 (No Poverty) or SDG6 (Clean Water and Sanitation) consistent with the national level, with the remaining 7 provinces varying. These common but differentiated SDG priorities reflect that to ensure a coordinated national response, China should pay more attention to the provincial situation, so that provincial governments can formulate more targeted policies in line with their own priorities towards accelerating sustainable development.

The 2030 Agenda for Sustainable Development, consisting of the 17 Sustainable Development Goals (SDGs) and 169 targets, is a global agenda for people, the planet, and prosperity to lead the world onto a sustainable and resilient path[1]. However, the SDGs have had a limited transformative impact so far[2,3]. One reason for this failure of SDGs is their selective implementation without considering their complex interactions[4]. As a system of interacting components, SDGs have complex interconnections with synergies (a pair of SDGs improve or deteriorate together) and trade-offs (one SDG improves while the other deteriorates), which play essential roles in achieving or inhibiting their effectiveness[4–6]. These complex interactions largely depend on the strategies applied to achieve an SDG. For example, infrastructure

[1]International Research Center of Big Data for Sustainable Development Goals, 100094 Beijing, China. [2]Key Laboratory of Digital Earth Science, Aerospace Information Research Institute, Chinese Academy of Sciences, 100094 Beijing, China. [3]The Key Laboratory of Land Surface Pattern and Simulation, Institute of Geographical Sciences and Natural Resources Research, Chinese Academy of Sciences, Beijing, China. [4]University of Chinese Academy of Sciences, Beijing, China. [5]Center for Systems Integration and Sustainability, Department of Fisheries and Wildlife, Michigan State University, East Lansing, MI, USA. [6]Integrated Research on Energy, Environment and Society (IREES), Energy and Sustainability Research Institute Groningen (ESRIG), University of Groningen, Groningen 9747 AG, Netherlands. [7]Potsdam Institute for Climate Impact Research (PIK), Member of the Leibniz Association, 14473 Potsdam, Germany. [8]School of Life and Environmental Sciences, Deakin University, Burwood, Victoria, Australia. [9]Luxembourg Institute of Science and Technology, Belvaux, Luxembourg. [10]Department of Biological Sciences, National University of Singapore, Singapore, Republic of Singapore. [11]School of Earth, Environment & Society, McMaster University, Hamilton, ON, Canada. [12]College of Water Resources and Civil Engineering, China Agricultural University, Beijing, China. [13]Luxembourg Institute of Socio-Economic Research, Maison des Sciences Humaines, 11, Porte des Sciences, L-4366 Esch-sur-Alzette/Belval, Luxembourg. [14]Natural Capital Project, Stanford University, Stanford, CA, USA. [15]Department of Geography, the University of Hong Kong, Hong Kong, China. ✉e-mail: wucy@igsnrr.ac.cn; chenfang@radi.ac.cn

like roads is necessary for poverty alleviation (SDG1) and economic development (SDG8) but may be detrimental for coast (SDG14) and land ecosystems (SDG15)[7]. Thus, to rescue SDGs from failure, one essential ingredient is to understand their synergies and trade-offs for determining priorities and improving the balance and integrity of policies towards achieving the 2030 Agenda holistically[4,8].

Systems thinking and analysis to assess the complex interactions among all 17 SDGs is at the forefront of sustainability research[9]. Existing SDG studies qualitatively evaluate SDG interactions by literature review[10–12], expert rating[13–16] and text mining[17]. Model-based analyses have focused more on environmental SDGs and less on the socio-economic dimensions[18]. With public databases, some research used network analysis to quantitatively analyze the differences in SDGs interaction networks at global and national levels[19–23]. However, due to economic, social and environmental heterogeneity, SDGs interactions may vary at a local or sub-national level within a country. Understanding SDGs interactions at these sub-national levels is essential since it is where the SDGs are implemented[3,4,24–26]. Understanding variations in SDGs interaction networks at different spatial levels, especially at the sub-national level, remains a fundamental research gap, which is essential for identifying context- and location-specific strategies for an integrated SDG implementation, especially for big countries like China.

As a large developing country in geographic area and population, China has experienced rapid economic development over the past few decades. However, it has also faced social problems and environmental challenges while striving for rapid economic development[24,27,28]. For example, climate change has exerted persistent impacts on China's ecological environment and socioeconomic development and brought serious threats to its food, water, ecology, energy, and urban operation security, as well as people's safety and property[29]. China's carbon emissions have significantly increased by around 10 times over the past 50 years[30]. Gender equality plays an important role in improving productivity and reducing gender discrimination and violence to promote economic development and social progress[31]. However, the gender gap in labor force participation between men and women rose from 9% to almost 15% between the 1990s and 2020[32]. Further, studies highlighted the challenges and disparities in SDG progress within China, suggesting that the uneven progress among the 17 SDGs at the sub-national level is a significant challenge for China's sustainable development[33–38]. We fill the above-highlighted research gaps by addressing the following two questions from the SDGs' synergies and trade-offs perspectives. (1) what are the common SDG priorities among subnational and national levels? (2) How do these priorities differentiate for synergies and trade-offs?

To this end, we aim to address the research gaps by analyzing the synergies and trade-offs among the 17 SDGs in China at the national, provincial and regional levels. We collected as much data as possible to cover all 17 SDGs at the national and sub-national levels on a yearly basis from 2000 to 2020. In total, an annual dataset of 102 indicators were used in our analysis (see Methods). We built synergy and trade-off networks at the national and sub-national levels, respectively. The synergy and trade-off intensity were set to be the weighted edge and the hub score of the 17 goals were in the nodes in the networks (see Methods). We analyzed the hub score to determine which goal served as the central hub in the synergy and trade-off networks. The larger the hub score, the more important the node as the central hub in the networks was. We analyzed these variations at national, provincial and regional levels among the 17 goals. Our findings can provide essential knowledge and insights into the priority of the SDGs to accelerate their implementation holistically at different spatial levels in China.

## Results
### The SDGs priorities at the national level
At the national level, 1023 out of 5151 indicator pairs showed synergies, and 374 pairs showed trade-offs with the average ABS(R) (absolute value of Spearman Correlation coefficient R) of 0.95 and 0.94 (Bonferroni corrected $p < 0.05$ and ABS(R) > 0.6) (Fig. 1a, b for indicators, please see the spreadsheet named "National" in Supplementary Data 4 of "synergies and trade-offs" for more details on the different categories in Fig. 1a, Fig. 2a, c for goals). The average Ratio (ratio of the number of the selected indicator pairs out of the total number of all possible combinations among goals) was 0.69 and 0.31 for synergies and trade-offs (Fig. 2b, d). Overall, we found that China faced challenges on SDG13 (Climate Change Action), SDG5 (Gender Equality), SDG17 (Partnerships for the Goals), and SDG16 (Peace, Justice and Strong Institutions), which showed highest hub scores in the trade-off network (0.96, 1, 0.86 and 0.81) and lowest in the synergy network (0.16, 0.46, 0.52 and 0.65) (Fig. 3a, b). SDG12 (Responsible Consumption and Production) showed a comparable score between synergy (0.69) and trade-off (0.71). China achieved co-benefits on the other 12 goals with the score in synergies higher than trade-offs (Fig. 3a, b, please see supplementary text in SI for more details on the priorities of SDGs at the national level with source data in Supplementary Data 3).

China became the world's largest emitter of carbon dioxide (CO2) in 2006[30]. China slipped from 63rd position in 2006 to 106th in the global gender gap rankings among 153 countries in 2019[39]. These brought serious trade-offs in SDG13 (Climate Action) and SDG5 (Gender Equality) along with the rapid economic development. Given combatting climate change can reinforce all 17 SDGs[29]. Gender equality is an enabler and accelerator for all the SDGs[31], the most important is that China overall needs to take decisive actions to mitigate the negative impact from SDG13 (Climate Action) and SDG5 (Gender Equality). These findings address explicitly the common priorities of the SDGs in trade-off (SDG13 and SDG5) and synergy (SDG1 and SDG6) among the different spatial levels in China.

China's trade-offs in SDG13 (Climate Action) and SDG5 (Gender Equality) are at a high level, and it needs to increase efforts at the national level for top-level design. For SDG13 (Climate Action), China needs to strengthen the top-level design of carbon peak and carbon neutral, propose a systematic, all-round plan for every sector, lead all-round green transformation to combat climate change. On one side, it requires reductions in high carbon emissions from economic development (SDG8)[30], especially the industry (SDG9) and traditional energy sectors (SDG7)[30], and from agriculture (SDG2)[32]; on the other side, it requires reinforcing the carbon sink in China's terrestrial ecosystems (SDG15)[33]. Besides, it also needs reducing the threat to water supplies and sanitation services (SDG6)[34,35], quality education (SDG4)[39,40], and human health (SDG3)[41,42]. For SDG5 (Gender Equality), the central government should take the lead to strengthen the top-level design and launch a package of much stronger plans and measures in every sector in the short and long terms to promote the all-round development of women and girl. In particular, China should reduce gender gap in women's employment rate and wage (SDG8 and SDG9)[43], participation in decision-making (SDG5)[44], healthcare (SDG3)[45], rural and secondary education (SDG4)[44,46], poverty reduction (SDG1)[47], rights protection in agriculture, forestry and animal husbandry (SDG2 and SDG15)[48], water and sanitation (SDG6)[49], and building partnership (SDG17)[50].

We chose the SDG indicators with the most available data at the national and sub-national levels simultaneously in our study to ensure the reliability of the results. The data came from a variety of official statistical yearbooks. Each type of data was collected by the corresponding official national ministries and provincial counterparts and was the most authoritative data currently available. Faced with such large-scale data collection, there were indeed varying degrees of data missing problems across the country and in different provinces due to differences in data collection capabilities, local conditions, personnel, budgets, etc. At the national scale, none of the indicator show no data and the data integrity is overall good. The ratio of indicators covering more than 15 years accounted for over 94%. The indicator of 9.c.1 with

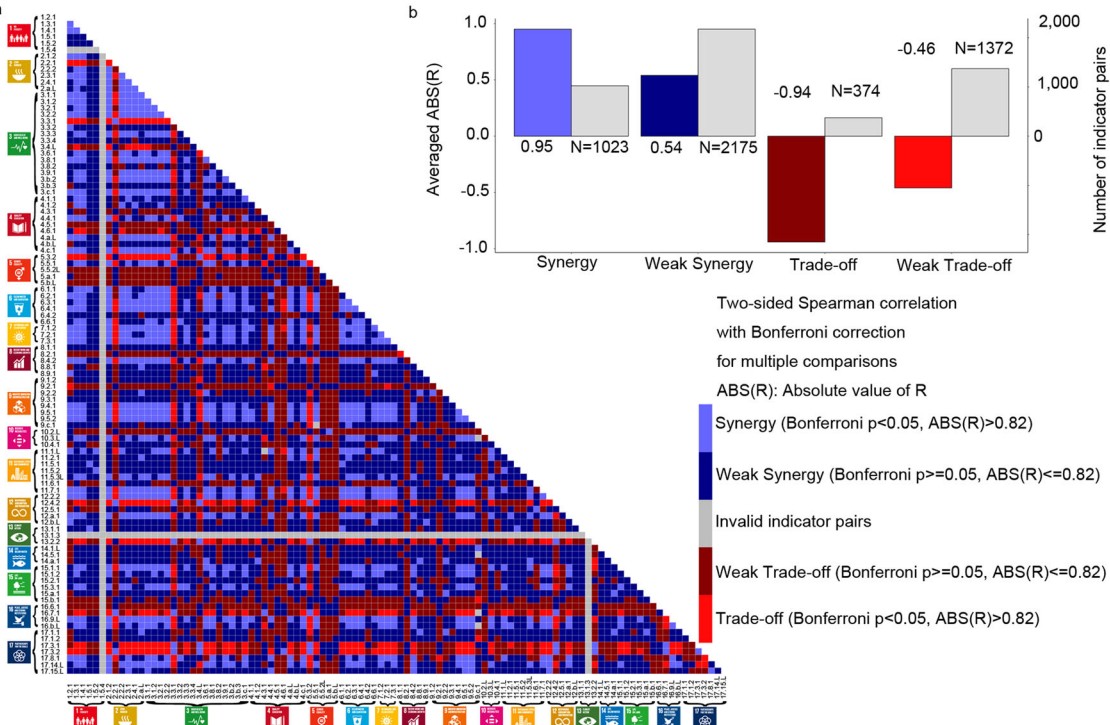

**Fig. 1 | The indicator pairs of synergies and trade-offs at the national level.**
**a** The distribution of the indicator pairs with numbering of the indicators showing the affiliation between the 102 indicators and 17 goals. The selection criteria are that Bonferroni- corrected *p* value are less than 0.05 and the absolute value of the Spearman correlation coefficient R (ABS(R)) are more than 0.6. Each indicator is judged to have a positive or negative impact on sustainable development based on its own meaning. The indicator pairs are divided in 5 groups, including synergy, trade-off, weak synergy, weak trade-off and invalid indicator pairs. Different colors indicate different SDGs following the official UN color palette. **b** The averaged R and number of indicator pairs for each group.

least data still covered 7 years. The lack of data in some years was mainly because the indicators were developed later and had not been collected by the official statistics before.

**The similarities and differences of SDGs priorities at provincial and regional levels**

At the provincial level, we found the differences in SDG synergies and trade-offs within China. In total, there were 244–872 pairs in synergies and 62–380 pairs in trade-offs with the averaged ABS(R) of 0.92–0.96 and 0.9–0.94 (Bonferroni corrected *p* < 0.05 and ABS(R) > 0.6) (see Supplementary Table 1 and Supplementary Data 4 of "Synergies and trade-offs" for more details on each province). At the national level, we could assess the overall situation across China. Among goals, SDG13 (Climate Action) and SDG5 (Gender Equality) had the lower hub scores in synergies on average (0.19 and 0.34) (Figs. 4a, c, 5a) and higher in trade-offs (0.76 for both) for 19 provinces consistent with the national level (Figs. 4b, d, 5b). 14 of these provinces had the highest trade-offs in SDG5 (Gender Equality), and 5 of them had the highest trade-offs in SDG13 (Climate Action) (see supplementary text of the results at the provincial level in SI for more details on the 19 provinces and Supplementary Fig. 2 for the trade-off networks at the provincial level with source data in Supplementary Data 1). The goal with the highest trade-offs differed among the other 12 provinces. They included SDG8 (Decent Work and Economic Growth) for Beijing and Chongqing, SDG10 (Reduced Inequalities) for Xinjiang, SDG3 (Good Health and Well-being) for Tibet, SDG9 (Industry, Innovation, and Infrastructure) for Heilongjiang, SDG12 (Responsible Consumption and Production) for Jilin, SDG17 (Partnerships for the Goals) for Guangdong, Shanghai, and Qinghai, SDG16 (Peace, Justice, and Strong Institutions) for Henan, and SDG2 (Zero Hunger) for Hubei and Tianjin (Fig. 4b, d), see supplementary Fig. 2 for the trade-off networks at the provincial level with source data in (Supplementary Data 1).

At the regional level, SDG13 (Climate Action) and SDG5 (Gender Equality) showed the highest trade-offs except for Northeast China, where SDG4 (Quality Education) had the most considerable trade-offs (Figs. 6b, 7c, d). The southern regions had a higher trade-off in SDG5 (Gender Equality) than the northern regions (Fig. 7c). However, SDG13 (Climate Action) had an opposite pattern between north and south in terms of trade-off (Fig. 7d).

We found that most of the SDGs had higher synergies than trade-offs. Among them, SDG1 (No Poverty) and SDG6 (Clean Water and Sanitation) showed high scores in synergies (0.98 and 0.97) (Figs. 4a, c, 5a) and low scores in trade-offs (0.35 and 0.42) for 24 provinces consistent with the national level. Among them, there were the highest synergies in SDG1 (No Poverty) and SDG6 (Clean Water and Sanitation) for 14 and 10 provinces (Figs. 4b, d, 5b, see supplementary text of the results at the provincial level in SI for more details on the 24 provinces and supplementary Fig. 1 for the synergy networks at the provincial level with source data in Supplementary Data 1). The goal with the highest synergies differed among the other 7 provinces. They included SDG2 for Henan, SDG11 (Sustainable Cities and Communities) for Inner Mongolia and Jilin, SDG7 (Affordable and Clean Energy) for Jiangxi, SDG4 (Quality Education) for Sichuan, Tibet, and Xinjiang (Fig. 4a, c, see supplementary Fig. 1 for the synergy networks at the provincial level with source data in Supplementary Data 1). In all six regions, SDG1 (No Poverty) and SDG6 (Clean Water and Sanitation) had the most synergies (Figs. 6a, 7a, b). These findings reflected the need for common but different priorities for SDGs in China at the national and provincial levels.

Each province or region had different synergy and trade-off priorities due to its own geographical location, resource endowment, climatic condition, topography, and historical development. The varied SDG priorities reflected that China should pay more attention to the actual situation of each province to ensure a coordinated national

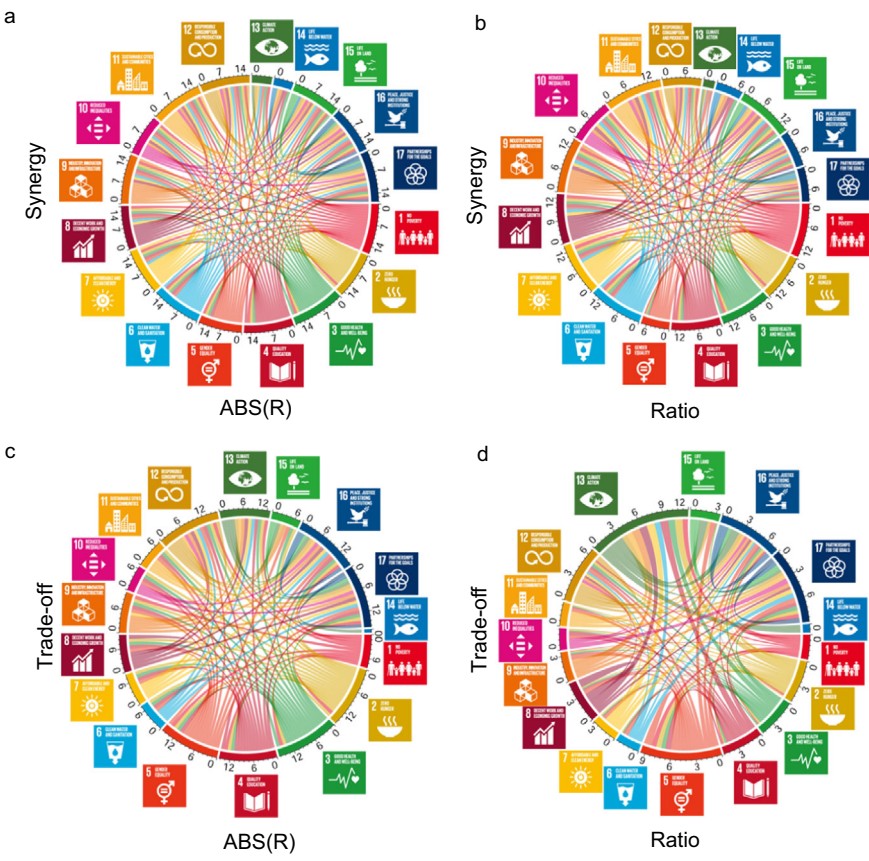

ABS(R): Absolute value of R

**Fig. 2 | The synergy and trade-off networks at the goal level. a, c** The absolute value of Spearman correlation coefficient R (ABS(R)) among goals at the national level. (**a**) is for synergy and (**c**) is for trade-off. Taking Fig. 2a as an example, the width of the colored line indicates the arithmetic mean of ABS(R) among goals calculated from the indicator pairs in Fig. 1a. The width of the arc represents the cumulative value of each line width for that goal. The number outside the circle is the scale of the goal. (**b**) and (**d**): the ratio of the number of the selected indicator pairs out of the total number of all possible combinations among goals (short for Ratio). **b** Is for synergy and (**d**) is for trade-off. Different colors indicate different SDGs following the official UN color palette.

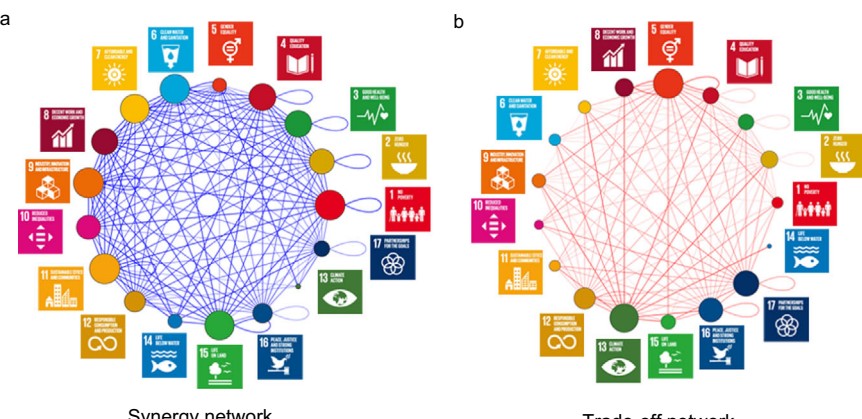

Synergy network Trade-off network

**Fig. 3 | The synergy and trade-off networks at the goal level built upon ABS(R) and Ratio.** The thickness of the edge in the network indicates the synergy or trade-off intensity among goals. The thicker the edge is the stronger the intensity is. The size of the circle suggests its importance as a central hub in the network. The larger the circle is, the more important the node as a central hub is. (**a**) is for synergy (hub score: 0.16–1) and (**b**) is for trade-off (hub score: 0.14-1). In the synergy network the edge is shown in blue and in red in the trade-off networks. Different colors indicate different SDGs following the official UN color palette.

response. In doing so, provincial governments could formulate more targeted policies aligning with regional and national SDG priorities towards accelerating sustainable development.

Here we took Tibet, which had the lowest GDP in China, as an example to discuss the provincial-level trade-offs. Restricted by the natural, geographical, climatic, and historical factors, Tibet's overall

medical and health services development was still lagging. The total medical and health resources were insufficient and unevenly distributed, and the medical service capabilities were weak. Problems still existed in institutional mechanisms, such as extensive management manner, insufficient strict implementation of the medical system, and insufficient procurement of much-needed drugs. From the perspective

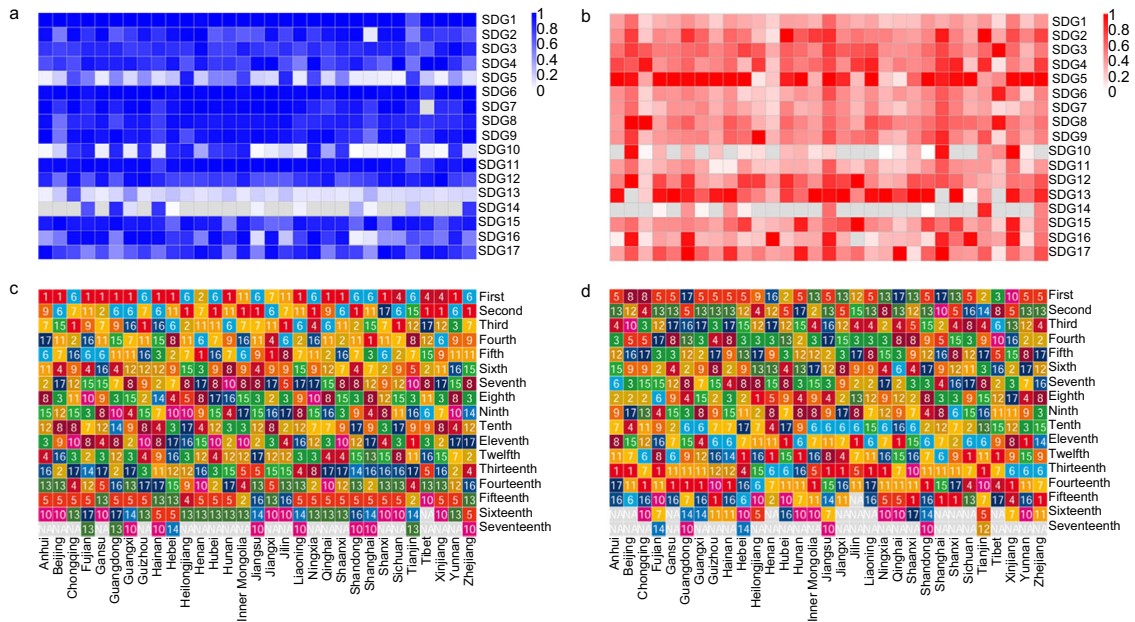

**Fig. 4 | The hubs scores of the 17 SDGs in synergy and trade-off networks at the provincial level.** The hub scores in the synergy (**a**) and trade-off (**b**) networks. The ranking of the 17 SDGs in the synergy (**c**) and trade-off (**d**) networks in order of its hub score. Different colors indicate different SDGs following the official UN color palette.

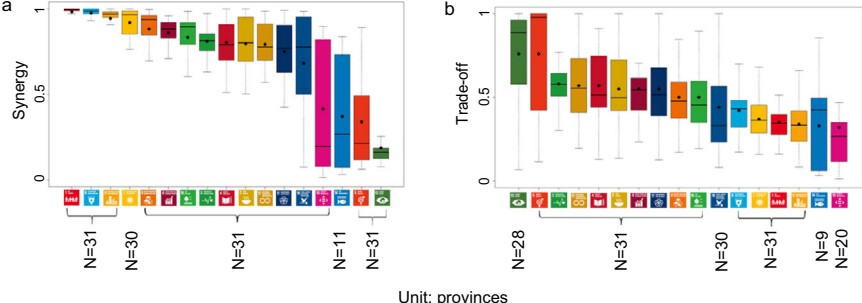

**Fig. 5 | The statistics of the hubs scores of the 17 SDGs in synergy and trade-off networks at the provincial level. a** Is for synergy and (**b**) is for trade-off. The black line in each box shows the minimum value, lower quartile, median, upper quartile and maximum value from left to right for each SDG. The solid black circle indicates the arithmetic mean value. Different colors indicate different SDGs following the official UN color palette.

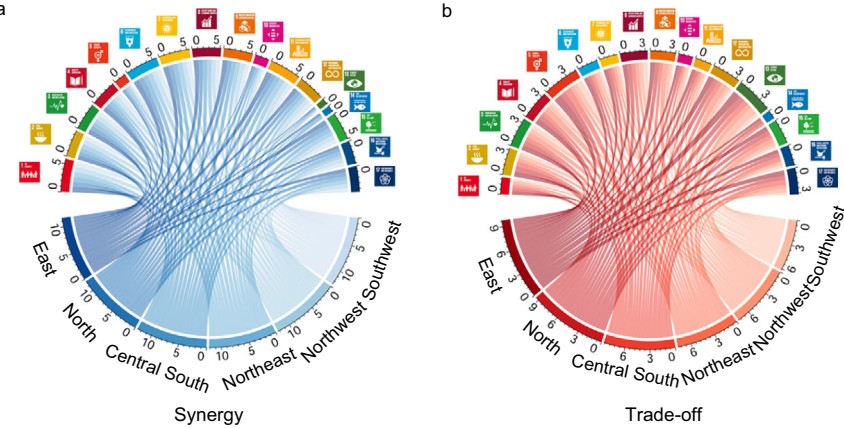

**Fig. 6 | The aggregated hub score of the 17 SDGs at the regional level. a** Is for synergy networks and (b) is for trade-off networks. Different colors indicate different SDGs following the official UN color palette.

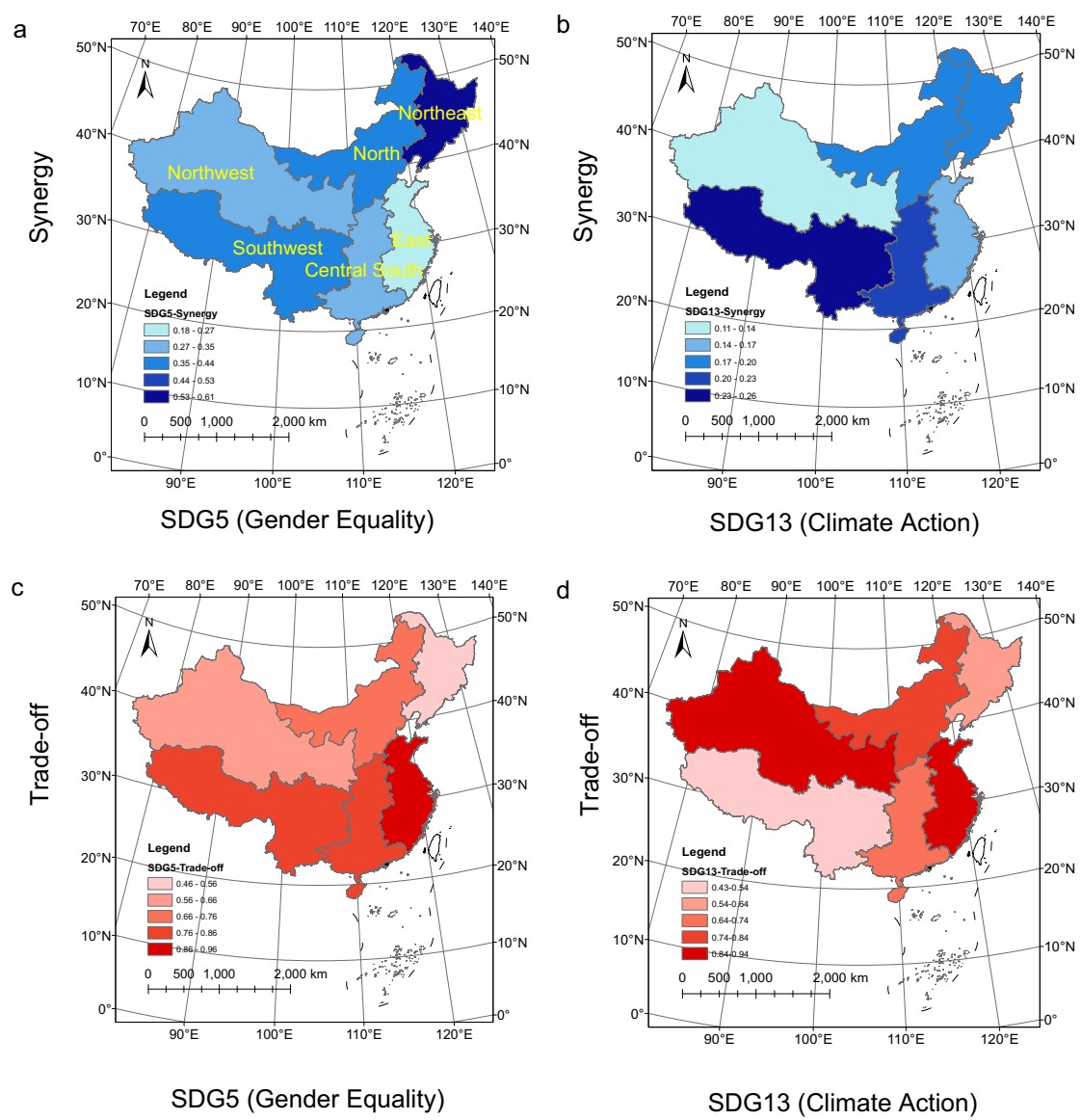

**Fig. 7 | The aggregated hub score of the 17 SDGs at the regional level.** The spatial pattern of the hub score of SDG5 (Gender equality) in the synergy (**a**) and trade-off (**c**) networks. The spatial pattern of the hub score of SDG13 (Climate action) in the synergy (**b**) and trade-off (**d**) networks.

of behavioral concept, the public's awareness of health was not strong, and the problem of diseases caused by unhealthy lifestyles was obvious[51]. Looking at the disease spectrum, the incidence rates of AIDS, tuberculosis, and hepatitis B per 100,000 people increased from 0.02, 75.07, and 52.62 in 2002 to 1.34, 150.13, and 107.29 in 2020, respectively[52]. The combined constraints of the natural, geographical, climatic, and historical factors made SDG3 (Good Health and Well-being) in Tibet have the highest trade-offs.

We took Guangdong as an example to discuss the provincial level synergies. Guangdong, at the forefront of reform and opening up, had the highest GDP in China. However, there was a large amount of poverty in the mountainous areas in the north and the underdeveloped areas in the west due to remote geographical location, numerous mountainous areas, insufficient transportation infrastructure, and single industrial structure. To this end, the Guangdong government integrated the targeted poverty alleviation campaign into the overall economic and social development plans for the overall planning, fully leveraging the synergies between SDG1 (No Poverty) and different SDGs. The government established a poverty reduction governance pattern emphasizing mutual promotion and focused on stimulating

the endogenous motivation to eliminate poverty. Through precise strategies, policies were implemented accurately and targeted to villages, households, and people. Different programs were implemented by promoting the combination of the market and the government leverages, including special poverty alleviation, industry poverty alleviation, and social poverty alleviation[53]. From 2013 to 2022, 2.5 million poor people in the province were lifted out of poverty, and the disposable income per capita increased by more than 2.6 times[54]. Through targeted poverty alleviation and poverty alleviation efforts, Guangdong was at the forefront of the country, making SDG1 (No Poverty) have the highest synergies. In the future, Guangdong needs to continue to leverage the synergistic advantages of SDG1 (No Poverty). Doing so will gradually narrow the income gap by further getting rid of relative poverty and realizing the rural revitalization.

Our results highlighted the need to prioritize different SDGs among Chinese provinces and regions based on understanding SDG interactions at various spatial levels. There were some common priorities, but the key SDGs differed at the provincial level, especially in the trade-off networks. Differentiated policies should be considered based on SDG interactions at the provincial and regional levels to maximize

synergies and mitigate trade-offs. For example, Beijing and Chongqing need to reduce the dominant trade-offs generated by their rapid economic development (SDG8)[55,56]. Xinjiang need to improve the inequalities (SDG10) at the social dimension[57]. Tibet had poor health conditions and need to mitigate the highly negative impact of good health and well-being (SDG3)[51]. Heilongjiang and Jilin faced high trade-offs from their traditional industries, the most important was to reduce the negative impact from industry (SDG9) and the consumption and production sectors (SDG12)[58].

At the provincial level, we collected data between 98 and 102 indicators for 30 provinces. For Tibet, with its remote location and difficult conditions, 94 indicators were collected to their maximum extent. After massive efforts on data collections, overall we compiled a set of 102 SDG indicators (118 original indicators and 102 after calculations), 81 targets, and 17 goals, at the provincial level on a yearly basis. This was greater than the number of indicators in the 2022 SDG Index and Dashboards Report (it used 88 indicators to assess China's SDG performances at the national level) and one previous study (it used 88 indicators, 71 targets, and 16 goals to calculate the score of 16 goals in China and analyze the SDGs interactions between 3 general categories based on the goal score)[59,60]. Our understanding of the SDGs interaction networks in China at provincial, regional, and national levels will evolve as more data becomes available.

## Discussion

We used social network analysis to quantitatively and systematically identify the priorities of the 17 SDGs through SDGs interaction networks using a unified dataset of 102 indicators at the sub-national and national levels in China. This understanding helps prioritize goals to implement the SDGs in an integrated and holistic way, so synergies can be reinforced and trade-offs can be mitigated.

The total carbon emissions in eastern China were significantly higher than those in central and western regions. The main reason was that the eastern region had a developed economy and a high demand for energy. For example, the rapid urbanization and industrial scale-up in the eastern region required a large amount of energy support, making the trade-off in SDG13 (Combating Climate Change) in Eastern China at a high level. The total carbon emissions in northwest China reached 1.61 billion tons in 2019, accounting for 15% of the country's total carbon emissions, with a growth rate of 5.8%[30]. The reason may be that in order to promote the economic development of the western region, China proposed a series of policies based on the "Western Development" strategy. The implementation of energy policies such as "West-to-east Electricity Transmission" and "West-to-east Gas Transmission" made the northwest region become China's energy important production base[61]. At the same time, some industries in the east were encouraged to move to the west, which to a certain extent strengthened the proportion of heavy and chemical industries in the west, which brought an increase in total carbon emissions in the west, making the trade-off in SDG13 (Combating Climate Change) in northwest China at a high level.

In recent years, the economic development rate of southern China has been significantly higher than that of northern China[28], which made the southern region, including East, Central South and Southwest China face more severe gender equality issues (SDG5). During China's planned economy stage (1952–1992), workers' employment allocation and salaries were determined by the Chinese government, which was engineered to result in a small gender wage gap. Yet with a market economy, corporations were given more autonomy, and women faced increased inequality in the workplace[62]. For instance, the gender gap in labor force participation between men and women nearly doubled over the last two decades, rising from 9% in the 1990s to almost 15% in 2020. The increase in gender earnings gap was even more pronounced[43].

Large countries such as China can take advantage of fiscal reallocation policies and administrative staffing mechanisms to limit trade-offs and leverage synergies. There was a long history of China using fiscal policy to move resources from more progressive to regions that lagging in socioeconomic development. Through the fiscal transfer payment policy, the central government transferred the fiscal surplus of high-GDP provinces such as Guangdong, Shanghai, and Beijing to Sichuan, Hunan, Hubei, Yunnan and other provinces to ensure education (SDG4), medical care (SDG3), balance the development of public services (SDG1) and narrow the gap on development among regions (SDG10)[63]. Our results indicate that the health goal in Tibet (SDG3) and the social inequality in Xinjiang (SDG10) has the greatest trade-offs, and are areas where traditional fiscal transfer payments need to be focused. The central government can also consider expanding the service scope of fiscal transfer payments. For example, in the provinces or regions where SDG13 (Combating Climate Change) dominates, fiscal transfer payment support can be considered to increase for Inner Mongolia, Jiangxi, Ningxia, Shaanxi, and Shanxi to address the challenge. In terms of SDG5 (Gender Equality), the increased support can occur in Anhui, Gansu, Guangxi, Guizhou, Hainan, Hebei, Hunan, Sichuan, and Yunnan, which have the most prominent trade-offs in SDG5 (Gender Equality), and are fiscally concerned with financial income less than expenditure. While Fujian, Liaoning, Shandong, Zhejiang, and Jiangsu can increase their support for SDG5 based on their own finances as they all have financial surplus.

Similarly, there is also the more potential to move local leaders to a region where there is a need for innovative thinking on the links between climate and gender in a modern democracy. Based on the performance highlights and comprehensive qualities of provincial leaders in the past, leaders can be moved to other provinces to promote learning from the experience of advanced provinces in dealing with SDGs in trade-offs and synergies[64]. For example, Tianjin has high synergies in SDG5 (Gender Equality), which is worthy of reference by all other provinces. Tibet with the highest trade-offs in SDG3 (Good Health and Well-being) can learn from the experience of Yunnan, which has greater synergy benefits in the health goal. Tianjin and Hubei having the highest trade-offs in SDG2 (Zero Hunger) can learn from Henan which obtain the highest synergies. Heilongjiang with the highest trade-offs in SDG9 (Industry, Innovation and Infrastructure) can learn from the experience of Anhui and Qinghai with high synergies in this goal.

Learning from our study, other countries worldwide could also make the common but differentiated SDGs priorities across spatial scales instead of cherry-picking. SDG13 (Climate Action) and SDG5 (Gender Equality) are the key hurdles for China to achieving 2030 agenda. They were also key goals for a successful implementation of SDGs globally[20,65]. Climate change continues to present a growing and significant global challenge to humanity and the biosphere in the 21st century[66]. Gender inequality is evident all over the world, with serious negative impacts on people's lives[67]. Promoting gender equality and climate actions will accelerate progress across the SDG systems[29,68]. The world can also learn from the compound positive impacts which China obtained by making progress on SDG1 (No Poverty) and SDG6 (Clean Water and Sanitation), particularly for low-income countries as eliminating poverty and ensuring proper water and sanitation facilities will contribute to accelerate progress across the SDG systems[68]. Besides, China has over three-decade long experiences on working on co-control between climate and air pollution that is increasingly focusing on synergies. This suggests the scope for applying the synergies logic more to the local level in China. Other countries can also learn from China's experiences on the synergic effect between climate and air pollution for their sustainable transformation[69,70].

To make the results meaningful in statistics, we used Bonferroni correction to avoid many possible spurious correlations as several statistical tests were being performed simultaneously[71]. We knew that the 17 goals, as the basic needs for humanity's survival and development on the planet, were related with each other and each indicator

reflected one aspect of the goal it belonged to[1-3]. We further used literature knowledge to explain the association between indicators for all the selected indicator pairs (see the supplementary file of explanations on the association between indicators for more details on each indicator pair). The network established in this article is an undirected weighted network. The Spearman Correlation used can detect the strength and positive and negative correlations between different indicators, but it cannot obtain directionality. The direction of the association involved causality, which is indeed a challenge. We have attempted to use Granger causality analysis on some closely related indicator pairs to check whether it can be used in our analysis to derive the direction of the association. The relationship between indicator 1.5.1 (Number of deaths, missing persons and directly affected persons attributed to disasters per persons attributed to disasters per 100,000 population) and indicator 1.5.2 (Direct economic loss attributed to disasters in relation to global gross domestic product (GDP)) at the national level were used as an example. Easily understanding, these two indicators were closely related with Spearman correlation coefficient of 0.82. However, Granger causality analysis showed that the P values of 1.5.1->1.5.2 ($P = 0.48$) and 1.5.2->1.5.1 ($P = 0.07$) were all higher than 0.05, indicating that it was not considered as Granger causality. Currently we found it was still difficult to reflect the causal relationship that they should exist. The current causal analysis still lacked of knowledge-based validation of the result. Still, the causal analysis often had drawbacks on short time series of data, and the control of the other variables in the system. Their effectiveness still needed to be tested further[21]. Additional data and the development of methods would enable us to move from correlation to reliable causality.

SDG synergies and trade-offs may be affected by cross-boundary interactions through flows of energy, people, technology, financial capital, etc[72]. As one example of a spillover effect within China, resource consumption by the more than 21 million residents in Beijing can exacerbate water scarcity (SDG6) and food insecurity (SDG2) in neighboring Hebei province and even the North China Plain since Beijing mainly relied on resources from its neighboring regions to support its development. A large part of the water and natural gas used in Beijing was provided through the "South-to-North Water Diversion" and "West-to-East Gas Transmission" Projects. Furthermore, the virtual resources such as water consumed in commodity production, such as food, clothes, etc. were transferred via interregional and international trade (SDG8 & SDG17), which allowed the receiving region to conserve local resources for other needs[73]. Future research and policy on SDGs interaction networks in China and other countries should account for cross-boundary issues.

With over two decades of data over China, we provide new insights into the common but differentiated SDGs priorities at provincial, regional, and national levels through interaction networks. In total, 19 provinces show the highest trade-offs in SDG13 (Combating Climate Change) or SDG5 (Gender Equality) consistent with the national level, with a difference rate of 12/31 while 24 provinces show the highest synergies in SDG1 (No Poverty) or SDG6 (Clean Water and Sanitation) consistent with the national level, with the difference rate of 7/31. These common but differentiated SDG priorities reflect that in the meantime of ensuring a coordinated national response, China should pay more attention to the actual situation of each province, so that provincial governments can formulate more targeted policies in line with provincial SDGs priorities towards high-quality sustainable development. Our study also provides China's example for determining priorities and improving the balance and integrity of measures towards achieving the SDGs to the other countries in the world.

## Methods

### Data collection and pre-processing

We selected indicators based on the definitions of goals, targets, and indicators in the UN official SDGs documents[74], the 2022 SDG Index and Dashboards Report from the Sustainable Development Solutions Network[59,75], and some recent studies[28,60]. For each SDG, we chose as many SDG indicators as feasible from the list of recommended indicators based on available data at the sub-national and national levels simultaneously and the availability of the indicators across the temporal scale. "The list of recommended indicators" referred to a number of documents, including the official SDG indicator from UN[74], the published references and reports[28,60,75], and the official statistical database of China, including the National Bureau of Statistics of the People's Republic of China, the China Statistical Yearbook[76], the Finance Yearbook of China[77], the China Statistical Yearbook on the Environment[78], the Educational Statistics Yearbook of China[79], the China Health Statistics Yearbook[52], the China Energy Statistical Yearbook[80], and other 9 Yearbooks from various ministries, such as insurance, urban construction, tourism, transportation & communications, industry, civil affairs, marine, forestry and population (Please see Supplementary Table 2 for a list of SDGs and their corresponding indicators, data sources, indicator sources and the maximum time period and see the Supplementary Data 5 for more details on the time period at the national and provincial levels). In the process of data collection, first of all, we selected indicators based on the official document of the Indicator of the United Nations with priority. If the indicators in the database can be directly matched or have similar meanings, we will directly select them with 40 indicators in total. Secondly, we supplemented the indicators that can be used based on the published literature and report with 51 indicators. Furthermore, according to the definition of target of SDGs, we selected indicator data in the database that can express the meaning with 11 indicators. Although each country has its own data collection status, our working flow on data collection can be applicable for other countries as well.

If the indicator had different elements, the average value of all the elements was calculated for the analysis. For example, the proportion of the population covered by insurance (endowment, unemployment, and medicare) (SDG 1, Indicator 1.3.1) was calculated from the average of that covered by endowment, unemployment, and medicare insurance. The averaged SDGs' indicators included the following: 1.3.1, 1.4.1, 2.3.1, 4.a.L, 4.c.1, 8.4.2, 9.1.2 and 12.2.2 (SI Tab. S2). We originally collected data for 118 indicators at national and sub-national levels annually. Then, the data were narrowed down to 102 indicators after the averaged calculation of various elements within one indicator. These data for 102 indicators are related to 81 targets and 17 goals.

### Synergy and trade-off calculation at the indicator level

The longitudinal Spearman correlation analyses covering non-linear relations were conducted between all 102 indicators at the 31 sub-national units one by one. The missing indicators data at certain years were dropped individually for each pairwise correlation by using the 'pairwise.complete.observation' mode. A Bonferroni correction was conducted to correct the P value when undertaking this many correlation tests[71]. The absolute value of the correlation coefficient |R| more than 0.6 were applied further to select the indicator pairs[5,60,81,82]. Since a higher value of an indicator did not necessarily mean a positive impact on sustainable development, we made a specific judgment based on the meaning of each indicator. For example, for the malnutrition rate of children under the age of 5 (SDG 2, Indicator 2.2.2), the lower value indicated a positive outcome. In contrast, for the proportion of GDP used to protect the biodiversity and ecosystem (SDG 15, Indicator 15.a.1), a lower value indicated a negative contribution to sustainable development. The detailed judgment table was listed in Supplementary Information, with "+1" indicating the better for sustainable development and "-1" indicating worse (see SI Table S1). We used literature knowledge to explain the association between indicators for all the selected indicator pairs (see Supplementary Data 2 for more details on the explanations of the associations).

## Synergy and trade-off calculation at the goal level

Based on the affiliation between indicator, target and goal[74], the synergy/trade-off intensity was calculated as follows:

$$\text{Intensity}_{type} = \text{Ratio}_{type} \times \text{ABS(R)}_{type} \tag{1}$$

$$\text{Ratio}_{type} = \frac{\text{EN}_{type}}{\text{TN}_{type}} \tag{2}$$

$$\text{ABS(R)}_{type} = \frac{\sum_{i=1}^{\text{EN}_{type}} \left| R_{type} \right|}{\text{EN}_{type}} \tag{3}$$

Where $Intensity_{type}$ was the synergy/trade-off intensity, *type* referred to synergy/trade-off, $Ratio_{type}$ was the the ratio of the number of the selected indicator pairs out of the total number of all possible combinations among goals, $ABS(R)_{type}$ was the the absolute value of Spearman correlation coefficient R among goals. $EN_{type}$ was the number of effective indicator pairs, $TN_{type}$ was the total number of indicator pairs between goals, $R_{type}$ was the Spearman correlation coefficient of the effective indicator pair. If we calculated the synergy and trade-off intensity directly at the goal level, we will ignore the fact that there were both synergies and trade-offs between different SDGs.

## Network analysis

Network analysis, which has been applied in social science[83], public heath[84], ecology[85] and biology[86] to study complex systems, is a holistic approach to studying the complexity of SDG interactions to identify the importance of goals or targets. The synergy and trade-off networks were built separately for the national and 31 provinces using iGraph package in R Studio, respectively[87].

We used the Hyperlink-Induced Topic Search (HITS) algorithm. This algorithm was proposed by Jon Kleinberg in 1999. It was originally used to sort web pages by importance and was later used for network analysis. HITS adopted the principle of mutual reinforcement. It was based on the following two easy-to-understand assumptions: (1) A high-quality authority node would be pointed to by many high-quality hub nodes. (2) A high-quality hub node would point to many high-quality authority nodes. Through multiple rounds of iterative calculations, each round of iterative calculations updated the two weights of each node until the weights were stable. The calculation formulas for the Authority value and Hub value of each node were as follows:

$$\text{auth}(p) = \sum_{q \in p_{to}} \text{hub}(q) \tag{4}$$

$$\text{hub}(p) = \sum_{q \in p_{from}} \text{auth}(q) \tag{5}$$

Among them, *p* was the target node, *q* was other nodes, $p_{to}$ represented the set of nodes pointed to point *p* from other nodes, and $p_{from}$ represented the set pointed to other nodes from point *p*. The algorithm process was as follows:

(1) Set the auth value and hub value of each node to 1;

(2) Use formulas to calculate and update the auth value of each node;

(3) Use the updated auth value to calculate and update the hub value of the node;

(4) Normalize the auth value and hub value;

(5) Repeat 2 ~ 4 until convergence or the stop iteration condition is reached.

For undirected matrices, the hub scores were the same as authority scores. The "hub_score" function from "igraph" package was used to calculate the hub score of each network in our analysis. For the further details of the algorithm, please refer to the paper from Kleinberg, 1999 in the reference[88].

The hub scores of the 17 SDGs were set as nodes, and the synergy or trade-off intensity among SDGs was set as the weighted edge in the network. These hub scores were used to calculate and assess the importance of the SDGs in the synergy and trade-off networks accounting for the direct and the indirect interactions. The larger the hub score was, the more important the node as a central hub was in the synergy or trade-off networks. The priority of the SDGs was identified based on the hub score in the networks from synergy and trade-off perspectives.

## The importance of the SDGs at different spatial levels

At the national level, we combined the 102 indicators in pairs, resulting in 5151 pairs in total. At the provincial level, the number of indicators ranges from 98 to 102 indicators for 30 provinces with the number of indicator pairs reaching between 4753 and 5151, except for Tibet having 94 indicators and 4371 pairs in total (for more details of each province, please refer to the data file in excel format). From indicator to goal, the importance of the SDGs was analyzed following the procedures above at the national and all the provincial levels. For the detailed statistics of the number of the selected indicator pairs and Spearman correlation coefficients of the 31 provinces, please refer to Supplementary Table 2 and the supplementary file of "Synergies and trade-offs" for further details. The results at the regional level were aggregated from those at provincial levels following the geographic regions divisions in China (See Supplementary Table 3 for more details).

## Reporting summary

Further information on research design is available in the Nature Portfolio Reporting Summary linked to this article.

# Data availability

The authors declare that the data supporting the findings of this study are available within the paper and its supplementary files. Source data are provided with this paper in Supplementary Data 1 and 3.

# Code availability

All R scripts and excel files used to process the data are available from the corresponding authors upon reasonable request.

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

## Acknowledgements

We thank Prof. Mark Stafford Smith from Commonwealth Scientific and Industrial Research Organization (CSIRO), Australia for providing comments on improving the quality of the paper. This study was supported by the Strategic Priority Research Program of the Chinese Academy of Sciences (XDA23100400), the National Natural Science Foundation of China grant (42125101), and National Key R&D Program of the Ministry of Science and Technology (2022YFC3800700). P. Pradhan acknowledges funding by the European Research Council (ERC) for the BeyondSDG project (Project number 101077492).

## Author contributions

C.Y.W. proposed the original idea. C.Y.W. and F.C. designed the research. Q.X. and C.Y.W. collected the data, performed the analysis and visualization, Q.X., C.Y.W., J.G.L. and P.P. wrote the original draft. J.L., P.P., B.B., F.C., T.S., L.R.C., A.G., Y.L., X.C., X.D., A.A., Y.L. and Z.X. reviewed, edited and revised the manuscript.

## Competing interests

The authors declare no competing interests.

## Additional information

**Peer review information** : *Nature Communications* thanks Mustafa Moinuddin, Anjal Prakash, and Eric Zusman for their contribution to the peer review of this work. A peer review file is available.

