## [Peer Review File · Nature Communications]

Intranational synergies and trade-offs reveal common and differentiated priorities of Sustainable Development Goals in ChinaReviewers' Comments:

Reviewer #1:

Remarks to the Author:

This is a useful paper where the authors attempted to address several major challenges facing the SDGs, especially at the subnational level: understanding how the SDGs are interconnected, how synergies and trade-offs work in this interconnected framework, and how to identify the priorities. The authors quantified the linkages among the SDG targets using national and subnational data for SDG indicators, which was then used for identifying synergies and trade-offs as well as for centrality analyses. Policy insights were then developed based on the results. Along with contributions to scientific knowledge on SDG linkages, the findings of the paper may also facilitate SDG policymaking at subnational and national levels. Some areas will need clarification and further elaboration. (see specific comments). The paper may need proofreading to correct some grammatical errors.

Specific comments

- The authors have pointed out that the existing research works on SDGs at the subnational levels have been limited primarily because of difficulties in data collection. In the current study, the authors have collected indicator-level time series data at national and provincial levels in China. What kind of difficulties were faced in collecting the data and how were those difficulties overcome? Many of the SDG indicators data may not be readily available (some SDG indicators are relatively new and some indicators are applicable only at the global level) and hence collecting data will require selecting an appropriate set of indicators (including proxies) first. The authors have also done that I guess (in line 394, the paper mentions "the list of recommended indicators", though it is not clear who recommended these indicators). It will be helpful if the paper explains the process of selecting these indicators. Did they follow any protocol or steps to select these indicators? And if a protocol has been developed, is it applicable to other countries as well?

- The authors mention (lines 428-429) that expert knowledge was used to explain the association between the indicators, but there is no explanation on who these experts are, how many experts were consulted and the process of receiving the expert knowledge/opinion. The supplementary information contains an explanation of the associations and references to literature but not about the experts or the consultation process.

- Are the association between indicators sometimes unidirectional and sometimes bidirectional? It seems so. The explanation column shows that most of them are bidirectional but there are a few instances (for example indicators 3.b.2 and 6.1.1) that are unidirectional. It may be a good idea to add a column to indicate if the relationship is uni- or bidirectional. As for the explanations, I felt that sometimes the bidirectional association is a bit imposed or a bit too indirect (e.g. between indicators 6.6.1 and 9.5.1).

- Related to the previous point, since the authors are using monotonic relationship between the indicators, I am wondering if the direction of association is also statistically derived from the analysis. Comparing this with the assumptions in the association table and among the provinces can offer policy insights.

- I think the literature review in this paper could be improved. There are existing studies, techniques and tools that have looked into SDG interactions based on causal association, correlation, synergies and trade-offs analysis as well as SNA-based centrality analyses (for example, Zhou and Moinuddin 2017, Weitz et al. 2018, Zhou, Moinuddin and Li, 2021; Allen, Metternicht and Wiedmann, 2019; Miola, Borhardt and Neher, 2019, and Jaramillo et al., 2019). Some of these studies have been cited in this paper. But, aside from this paper's focus on multiple spatial levels, are there other methodological differences and advantages compared to these other studies? The approach to analyse goal-level synergies and trade-offs may merit highlighting here. Furthermore, there are some excellent studies that have explored the spatial variability of SDGs, including but not limited to China.

For example, Wang et al. (2020) evaluated SDGs in the Chinese provinces. While the approach may be different, some of these studies should be referred to in this paper to indicate novelty and value-added.

- The explanation of synergies/trade-offs analysis at the provincial level (page 9) is a bit confusing. Why are indicator pairs presented as an aggregate of all 31 provinces (158,100 indicator pairs) instead of individually 5,100 pairs for each province? Then again on the explanation of networks (page 24), it says the networks were built separately (though at the goal level?).

- The explanation of network analysis methodology (page 24) is a bit vague. Maybe the authors can add more on the centralities and their denotations (including the choice of hub/authority centralities) and the overall process.

- One additional observation: On page 3 (and in the abstract), the authors hold that understanding the synergies and trade-offs for determining the SDG priorities is essential "to rescue the SDGs from failing". However, there are several other critical reasons – such as securing financing (particularly for developing countries) – that are equally (if not more) important for the effective implementation of the SDGs. Halfway through the implementation of the SDGs with no region or country of the world being on track in achieving the SDGs, it may be realistic not to suggest that one single approach will be effective in rescuing the SDGs.

Reviewer #2:

Remarks to the Author:

The paper provides a provocatively titled analysis of the potential impacts of gender equity and climate on achieving the SDGs in China. It, moreover, offers a useful and technically sound analysis of the synergies and trade-offs between the SDGs at the local and national levels in China. However, the article has some significant problems in my view in its framing, presentation of results, and policy implications. To be considered for publication, major revisions are needed.

1) Framing: Consider a new title that emphasizes the local level interlinkages analysis and its implications in China. In my view, the paper does not really focus as much on gender and climate interactions as offering a local level intranational comparison of synergies and trade-offs in China. That is the data gap highlighted in the paper but it is not featured in the title.

2) Presenting Results: Consider a more focused and detailed explanation of what are the results for different regions; and what the intranational variation implies for a larger country like China. The current presentation is rather scatter shot with Tibet and Xinjiang having this kind of trade-off but not much discussion of the context and the reasons that such trade-offs arise. A similar claim could be made the synergies arguments that, in my view, lack the context and depth one might desire. To address this, I would suggest one or two case studies that go into greater depth on provincial level synergies and trade-offs.

3) Policy Implications: Here again, I found the discussion of policy thin and weak. I would suggest a greater emphasis on underlining how large countries such as China can take advantage of fiscal reallocation policies and administrative staffing mechanisms to limit trade-offs and leverage synergies. For example, there is a long history of China using fiscal policy to move resources from more progressive to regions that lagging in socioeconomic development. Similarly, there is also the potential—more so that in a modern democracy—to move local leaders to a region where there is a need for innovative thinking on the links between climate and gender. Please check the China studies literature to offer some suggestions that consider these possibilities. My fear, otherwise, is that the article is based chiefly on a modelling analysis that lacks the empirical richness needed to make it really useful for China and other countries seeking to balance competing priorities.

Reviewer #3:

Remarks to the Author:

Review Report: Climate action and gender equality matter most for China's sustainable development

While the paper offers valuable insights into the complex interactions among SDGs in China, there are areas in each section that can be improved to enhance the overall impact and utility of the research. Addressing these areas for improvement will contribute to a more robust and comprehensive research paper. Overall, the abstract and introduction provide a good starting point for the paper but can be improved by including more specific findings and implications and enhancing their engagement with the reader. Additionally, providing more context and citations could further support the research's significance. The Results section provides valuable quantitative insights into the priority of SDGs at the national level in China. To enhance this section, the authors should provide more detailed explanations of metrics, offer a deeper interpretation of results, discuss data quality, and make explicit connections to the research questions and implications for policy and practice. The "Discussion and Policy Implication" section effectively outlines the policy implications of the research findings, emphasising the importance of addressing climate action and gender equality while harnessing the positive impacts of poverty alleviation and clean water initiatives. To enhance this section, the authors should provide a deeper analysis, explore cross-boundary interactions, connect findings to the global context, offer specific policy recommendations, and conclude with a concise summary of key takeaways. Some specific points in these sections are slated below.

1. The paper effectively communicates its research objectives and the identified research gap in the abstract section. It excels in clarity, making the central focus of the research readily understandable. However, there are areas where improvement is needed, such as the absence of specific findings and quantitative data, the lack of clear implications, the need for specific keywords to enhance discoverability, and the potential for a bit more detail to convey key insights effectively. Kindly consider some of these points:

- While it mentions identifying climate change (SDG13) and gender equality (SDG5) as key hurdles, it lacks specific findings and quantitative data to make the abstract more informative and enticing to potential readers.
- The abstract does not clearly indicate why these findings are important or what implications they might have for China's sustainable development.
- Including specific keywords related to climate change, gender equality, and SDGs could enhance the discoverability of the paper.
- While being concise is important, the abstract could benefit from more detail, particularly in summarising key findings or insights.

2. The introduction provides a solid foundation for the paper, setting the stage by introducing the 2030 Agenda and justifying China's selection as a case study. It also clearly identifies the research gap related to SDG interactions. However, it could benefit from more explicit relevance of gender and climate goals to China's sustainable development, a smoother transition to the methodology section, a more engaging introduction, and more comprehensive citations to support its arguments. Some points to consider:

a. Explicit Relevance of Gender and Climate: While the introduction mentions that combating climate change (SDG13) and improving gender equality (SDG5) are key hurdles, it could be more explicit about why these specific goals are relevant to China's sustainable development.

b. Transition to Methodology: The transition from the research gap to the methodology section could be smoother. It would be helpful to briefly introduce how the study aims to address the research gap before diving into the methodology.

c. Engagement of the Reader: The introduction could be more engaging by using more vivid language or a compelling statistic to draw the reader's attention to the importance of the research.

d. Citations: It could benefit from citing specific studies or reports highlighting the challenges and disparities in SDG progress within China. This would strengthen the foundation of the research.

3. Results: The results section offers valuable quantitative insights into the priority of SDGs at the national level in China. Strengths include quantitative rigour, effective visualisation, clear identification

of key goals, and a comparison between synergy and trade-off networks. Areas for improvement include a more detailed explanation of metrics, a deeper interpretation of results, a discussion of data quality and reliability, references to supplementary information, and a stronger connection to research questions. The section analyses the priority of Sustainable Development Goals (SDGs) at the national level in China based on synergy and trade-off networks. Some points to consider:

- a. Detailed Explanation of Metrics: While the section mentions ABS(R) and the Ratio, it could benefit from a more detailed explanation of these metrics and their significance. Readers may not be familiar with these measures, and providing a brief description or formula would enhance clarity.
- b. Interpretation of Results: The section could improve by providing more interpretation. For example, why do SDG13 and SDG5 have the highest hub scores in the trade-off network? What are the practical implications of these findings for China's sustainable development? A deeper analysis of the "why" behind these results would strengthen the section.
- c. Data Quality and Reliability: Given the extensive dataset used for analysis, it would be helpful to briefly discuss the quality and reliability of the data. Were there any challenges or limitations in data collection that could affect the results?
- d. References to Supplementary Information: The section references supplementary information (SI) for more details. However, it could be improved by briefly summarising the key findings or insights from the supplementary information, making it more accessible to readers.
- e. Connection to Research Questions: While the section discusses findings related to SDG priorities, it could be more explicit in connecting these findings to the research questions posed in the introduction. How do these findings address common priorities among different spatial levels in China?

4. Section - Similarity and Differences of SDG Priority among Provinces: This section analyses SDG priorities at the provincial level in China, presenting quantitative data and comparisons across goals. Strengths include quantitative rigour, effective visualisation, regional analysis, and comparisons. However, it could benefit from a deeper interpretation of results, a stronger connection to research questions, a discussion of data quality, integration with previous findings, and an exploration of policy implications for regional development. The section provides valuable quantitative insights into SDG priorities in China's provincial level, focusing on synergy and trade-off networks. To enhance this section, the authors should provide a deeper interpretation of results, connect findings to research questions, discuss data quality, integrate findings with previous results, and explore policy implications for regional development.

- a. Interpretation of Results: While the section presents the data and identifies differences in SDG priorities, it could benefit from a deeper interpretation of the results. Why do certain provinces or regions prioritise specific SDGs over others? What are the implications of these variations for policy and development strategies?
- b. Relevance to Research Questions: The section should explicitly connect the findings to the research questions in the introduction. How do these provincial variations in SDG priorities relate to the overall research objectives of understanding SDG interactions at different spatial levels?
- c. Data Quality: Similar to the earlier part of the Results section, it would be beneficial to briefly discuss the quality and reliability of the data used for provincial-level analysis. This can help readers understand the potential limitations of the findings.
- d. Integration with Previous Findings: Integrating the provincial-level findings with the national-level findings discussed earlier in the Results section is essential. Are there consistencies or differences between the two levels of analysis? How do these findings contribute to understanding China's SDG priorities?
- e. Policy Implications: The section lacks discussion of the practical implications of the provincial-level findings for policymakers and development practitioners. A brief discussion of how these insights can inform regional development strategies or SDG implementation plans would be valuable.

5. Section: Discussion and Policy Implication: The discussion section interprets findings and discusses policy implications, focusing on policy recommendations. Strengths include clear policy implications, use of real-world examples, recognition of regional differentiation, and explanation of statistical rigour. Areas for improvement include a need for deeper analysis, exploration of cross-boundary interactions, better connection to the broader global context, more specific policy recommendations, and concise

concluding remarks.

- a. While the section discusses SDGs' common and differentiated priorities, it could benefit from a deeper analysis of why certain goals are more challenging or have higher trade-offs in specific regions. Exploring the underlying causes and context-specific factors would enhance the discussion.
- b. The section mentions that SDG synergies and trade-offs may be affected by cross-boundary interactions but does not delve into this aspect in detail. Expanding on how these cross-boundary interactions influence SDG priorities would enrich the discussion.
- c. The discussion could be improved by connecting the findings to the broader context of sustainable development and global SDG implementation. How do China's experiences and priorities compare to those of other countries facing similar challenges?
- d. While the section identifies key challenges and priorities, it could provide more specific policy recommendations or strategies for addressing them. Offering actionable guidance for policymakers would enhance the section's practical utility.
- e. The section could benefit from a concise summary or concluding remarks that reiterate the main policy implications and the significance of the research.

Response to the Editor and Reviewers

We would like to express our gratitude to the editor and anonymous reviewers for their valuable comments and suggestions for improving the quality of the paper. We have carefully considered all the points raised by them. We are providing detailed point-by-point responses to all questions and recommendations by the reviewers. In the responses below, red fonts are the revised texts with the column number in the revised version.

Part 1: Response to reviewer 1

This is a useful paper where the authors attempted to address several major challenges facing the SDGs, especially at the subnational level: understanding how the SDGs are interconnected, how synergies and trade-offs work in this interconnected framework, and how to identify the priorities. The authors quantified the linkages among the SDG targets using national and subnational data for SDG indicators, which was then used for identifying synergies and trade-offs as well as for centrality analyses. Policy insights were then developed based on the results. Along with contributions to scientific knowledge on SDG linkages, the findings of the paper may also facilitate SDG policymaking at subnational and national levels. Some areas will need clarification and further elaboration. (see specific comments). The paper may need proofreading to correct some grammatical errors.

Response: Thanks for the comments. We have made more clear clarifications and elaborations on the specific comments and made a thorough proofreading.

Specific comments

Issue 1: The authors have pointed out that the existing research works on SDGs at the subnational levels have been limited primarily because of difficulties in data collection. In the current study, the authors have collected indicator-level time series data at national and provincial levels in China. What kind of difficulties were faced in collecting the data and how were those difficulties overcome? Many of the SDG indicators data may not be readily available (some SDG indicators are relatively new and some indicators are applicable only at the global level) and hence collecting data will require selecting an appropriate set of indicators (including proxies) first. The authors have also done that I guess (in line 394, the paper mentions “the list of recommended indicators”, though it is not clear who recommended these indicators). It will be helpful if the paper explains the process of selecting these indicators. Did they follow any protocol or steps to select these indicators? And if a protocol has been developed, is it applicable to other countries as well?

Response: Revised as suggested.

Thanks for the comments. We have added the information on selecting indicators as follows (line 460-line 477 and we also added one column of indicator sources in the sixth column in SI Table 2):

“The list of recommended indicators” referred to a number of documents, including the official SDG indicator from UN (Inter-agency and Expert Group on SDG Indicators, 2023), the published references and reports (Xu et al., 2020; Zhang et al., 2022; Sustainable Development Solutions Network, 2015), and the official statistical database of China, including the National Bureau of Statistics of the People’s Republic of China, the China Statistical Yearbook (China Statistics Press, 2001–2021), the Finance Yearbook of China (China Financial & Economic Publishing House, 2001-2021), the China Statistical Yearbook on the Environment (China Statistical Yearbook on Environment, 2001-2021), the Educational Statistics Yearbook of China (People's Education Press, 2001-2021), the China Health Statistics Yearbook (China Health and Family Planning Statistical Yearbook, 2001-2021), the China Energy Statistical Yearbook (China Energy Statistical Yearbook, 2001-2021), and other 9 Yearbooks from various ministries, such as insurance, urban construction, tourism, transportation & communications, industry, civil affairs, marine, forestry and population (Please see SI Table 2 for a list of SDGs and their corresponding indicators, data sources, indicator sources and the maximum time period and see the supplementary data file for more details on the time period at the national and provincial levels). In the process of data collection, first of all, we selected indicators based on the official document of the Indicator of the United Nations with priority. If the indicators in the database can be directly matched or have similar meanings, we will directly select them with 40 indicators in total. Secondly, we supplemented the indicators that can be used based on the published literature and report with 51 indicators. Furthermore, according to the definition of target of SDGs, we selected indicator data in the database that can express the meaning with 11 indicators. Although each country has its own data collection status, our working flow on data collection can be applicable for other countries as well.

Inter-agency and Expert Group on SDG Indicators, (IAEG-SDGs), Tier Classification for Global SDG Indicators. Accessed at: https://unstats.un.org/sdgs/files/Tier%20Classification%20of%20SDG%20Indicators_31%20Mar%202023_web.pdf (2023).

Xu Z., et al. Assessing progress towards sustainable development over space and time. *Nature* 577, 74–78, <https://doi.org/10.1038/s41586-019-1846-3> (2020).

Zhang J, Wang S, Pradhan P, et al. Untangling the interactions among the Sustainable Development Goals in China. *Sci. Bull.* 67(9), 977-984 (2022).

Sustainable Development Solutions Network, Indicators and a Monitoring Framework for the Sustainable Development Goals: Launching a Data Revolution for the SDGs (2015).

China Statistics Press, “China Statistical Yearbook” (National Bureau of Statistics of the People’s Republic of China, 2001–2021) [in Chinese].

China Financial & Economic Publishing House, “Finance Yearbook of China” (Ministry of Finance of the People's Republic of China, 2001-2021) [in Chinese].

China Statistics Press, “China Statistical Yearbook on Environment” (National Bureau of Statistics & State Environmental Protection Administration of the People's Republic of China, 2001-2021) [in Chinese].

People's Education Press, “Educational Statistics Yearbook of China” (Ministry of Education of the People's Republic of China, 2001-2021) [in Chinese].

China Statistics Press, “China Health and Family Planning Statistical Yearbook” (National Bureau of Statistics of the People's Republic of China, 2001-2021) [in Chinese].

China Statistics Press, “China Energy Statistical Yearbook” (National Bureau of Statistics of the People's Republic of China, 2001-2021) [in Chinese].”

Issue 2: The authors mention (lines 428-429) that expert knowledge was used to explain the association between the indicators, but there is no explanation on who these experts are, how many experts were consulted and the process of receiving the expert knowledge/opinion. The supplementary information contains an explanation of the associations and references to literature but not about the experts or the consultation process.

Response: Revised as suggested.

Sorry for the misunderstanding. This is a wording error. What we originally wanted to express was the expert knowledge in the literature. The revised version is shown as follows (line 389 and line 501):

“literature knowledge”

Issue 3: Are the association between indicators sometimes unidirectional and sometimes bidirectional? It seems so. The explanation column shows that most of them are bidirectional but there are a few instances (for example indicators 3.b.2 and 6.1.1) that are unidirectional. It may be a good idea to add a column to indicate if the relationship is uni- or bidirectional. As for the explanations, I felt that sometimes the bidirectional association is a bit imposed or a bit too indirect (e.g. between indicators 6.6.1 and 9.5.1).

Response: Revised as suggested.

Thanks for the comments. We provide this explanation table to find the basis for the connection between indicators based on literature. Indeed, some indicators have indirect relationships. In response to the comment, we rechecked each explanation one by one, added one column to indicate if the relationship is unidirectional or bidirectional, and modified some explanations and literature marked in yellow in the excel file. If the bidirectional association is a bit imposed or a bit too indirect, we have changed them into unidirectional to make it more understandable. Please see the revised supplementary file of “Explanations on the association between indicators_R1”.

Issue 4: Related to the previous point, since the authors are using monotonic relationship between the indicators, I am wondering if the direction of association is also statistically derived from the analysis. Comparing this with the assumptions in the association table and among the provinces can offer policy insights.

Response: provide explanations and explorative experiment results

Thanks for your comment. The network established in this article is an undirected weighted network. The Spearman Correlation used can detect the strength and positive and negative correlations between different indicators, but it cannot obtain directionality. The direction of the association involves causality, which is indeed a challenge. In response to the comment, we have attempted to use Granger causality analysis on some closely related indicator pairs to check whether it can be used in our analysis to derive the direction of the association. The relationship between indicator 1.5.1 (Number of deaths, missing persons and directly affected persons attributed to disasters per persons attributed to disasters per 100,000 population) and indicator 1.5.2 (Direct economic loss attributed to disasters in relation to global gross domestic product (GDP)) at the national level was used as an example. Easily understood, these two indicators are closely related with Spearman correlation coefficient being 0.82. However, Granger causality analysis shows that: the P values of 1.5.1->1.5.2 (P=0.48) and 1.5.2->1.5.1 (P=0.07) are all higher than 0.05, indicating that it is not considered as Granger causality. We opt to put this as limitations in the discussions as follows (line 391-line 408):

“The network established in this article is an undirected weighted network. The Spearman Correlation used can detect the strength and positive and negative correlations between different indicators, but it cannot obtain directionality. The direction of the association involved causality, which is indeed a challenge. We have attempted to use Granger causality analysis on some closely related indicator pairs to check whether it can be used in our analysis to derive the direction of the association. The relationship between indicator 1.5.1 (Number of deaths, missing persons and directly affected persons attributed to disasters per persons attributed to disasters per 100,000 population) and indicator 1.5.2 (Direct economic loss attributed to disasters in relation to global gross domestic product (GDP)) at the national level were used as an example. Easily understanding, these two indicators were closely related with Spearman correlation coefficient of 0.82. However, Granger causality analysis showed that the P values of 1.5.1->1.5.2 (P=0.48) and 1.5.2->1.5.1 (P=0.07) were all higher than 0.05, indicating that it was not considered as Granger causality. Currently we found it was still difficult to reflect the causal relationship that they should exist. The current causal analysis still lacked of knowledge-based validation of the result. Still, the causal analysis often had drawbacks on short time series of data, and the control of the other variables in the system. Their effectiveness still needed to be tested further. Additional data and the development of methods would enable us to move from correlation to reliable causality.”

Attached is the program running result:

```
##### Granger causality test: 1.5.1->1.5.2#####  
> grangertest(Test1$`1.5.1` ~ Test1$`1.5.2`, order = 1, data = Test1)  
Granger causality test  
Model 1: Test1$`1.5.1` ~ Lags(Test1$`1.5.1`, 1:1) + Lags(Test1$`1.5.2`, 1:1)  
Model 2: Test1$`1.5.1` ~ Lags(Test1$`1.5.1`, 1:1)  
Res.Df Df    F Pr(>F)  
1    13
```

```
2 14 -1 0.5327 0.4784
```

```
##### Granger causality test: 1.5.2->1.5.1#####
```

```
> grangertest(Test1$`1.5.2` ~ Test1$`1.5.1`, order = 1, data = Test1)
```

```
Granger causality test
```

```
Model 1: Test1$`1.5.2` ~ Lags(Test1$`1.5.2`, 1:1) + Lags(Test1$`1.5.1`, 1:1)
```

```
Model 2: Test1$`1.5.2` ~ Lags(Test1$`1.5.2`, 1:1)
```

```
Res.Df Df F Pr(>F)
```

```
1 13
```

```
2 14 -1 3.9753 0.06758 .
```

```
---
```

```
Signif. codes: 0 '***' 0.001 '**' 0.01 '*' 0.05 '.' 0.1 ' ' 1
```

```
##### Spearman correlation between 1.5.1 and 1.5.2#####
```

```
> cor.test(Test1$`1.5.1`, Test1$`1.5.2`, method = "spearman")
```

```
Spearman's rank correlation rho
```

```
data: Test1$`1.5.1` and Test1$`1.5.2`
```

```
S = 146, p-value = 6.004e-05
```

```
alternative hypothesis: true rho is not equal to 0
```

```
sample estimates:
```

```
rho
```

```
0.8210784
```

```
##### Spearman correlation between 1.5.2 and 1.5.1#####
```

```
> cor.test(Test1$`1.5.2`, Test1$`1.5.1`, method = "spearman")
```

```
Spearman's rank correlation rho
```

```
data: Test1$`1.5.2` and Test1$`1.5.1`
```

```
S = 146, p-value = 6.004e-05
```

```
alternative hypothesis: true rho is not equal to 0
```

```
sample estimates:
```

```
rho
```

```
0.8210784
```

Issue 5: I think the literature review in this paper could be improved. There are existing studies, techniques and tools that have looked into SDG interactions based on causal association, correlation, synergies and trade-offs analysis as well as SNA-based centrality analyses (for example, Zhou and Moinuddin 2017, Weitz et al. 2018, Zhou, Moinuddin and Li, 2021; Allen, Metternicht and Wiedmann, 2019; Miola, Borchardt and Neher, 2019, and Jaramillo et al., 2019). Some of these studies have been cited in this paper. But, aside from this paper's focus on multiple spatial levels, are there other methodological differences and advantages compared to these other studies? The approach to analyse goal-level synergies and trade-offs may merit highlighting here. Furthermore, there are some excellent studies that have explored the spatial variability of SDGs, including but not limited to China. For example, Wang et al. (2020) evaluated SDGs in the Chinese provinces. While the approach may be different, some of these studies should be referred to in this paper to indicate novelty and value-added.

Response: Revised as suggested.

Thanks for the comments. We have added the following literature in Line 77-78 as follows:

“Existing SDG studies qualitatively evaluate SDG interactions by literature review (Miola et al., 2019; Jaramillo et al., 2019), expert rating (Weitz et al., 2018) and text mining.

Weitz, N., Carlsen, H., Nilsson, M., & Skånberg, K.. Towards systemic and contextual priority setting for implementing the 2030 Agenda. *Sustainability science*, 13, 531-548 (2018).

Miola, A., Borchardt, S., Neher, F., & Buscaglia, D.. Interlinkages and policy coherence for the Sustainable Development Goals implementation. The Joint Research Centre (JRC) (2019).

Jaramillo, F., Desormeaux, A., Hedlund, J., Jawitz, J. W., Clerici, N., Piemontese, L., ... & Åhlén, I.. Priorities and interactions of sustainable development goals (SDGs) with focus on wetlands. *Water*, 11(3), 619 (2019).”

We have added the following literature in Line 79-line 81 as follows:

“With public databases, some research used network analysis to quantitatively analyze the differences in SDGs interaction networks at global and national levels (Zhou et al., 2021; Allen et al., 2019).

Zhou X, Moinuddin M, Li Y SDG interlinkages analysis and visualisation tool (V3.0). Institute for Global Environmental Strategies (IGES), Hayama, Japan (2021). Available at: <https://sdginterlinkages.iges.jp/visualisationtool.html>

Allen, C., Metternicht, G., & Wiedmann, T.. Prioritising SDG targets: Assessing baselines, gaps and interlinkages. *Sustainability Science*, 14, 421-438 (2019).”

We have added the following literature in Line 79-line 81 as follows:

“Understanding SDGs interactions at these sub-national levels is essential since it is where the SDGs are implemented (Cole et al., 2017; Wang et al., 2020; de Miguel Ramos et al., 2020). Line: 83-84

Wang, Y., Lu, Y., He, G., Wang, C., Yuan, J., & Cao, X.. Spatial variability of sustainable development goals in China: A provincial level evaluation. *Environmental Development*, 35, 100483 (2020).

Cole, M. J., Bailey, R. M., & New, M. G.. Spatial variability in sustainable development trajectories in South Africa: provincial level safe and just operating spaces. *Sustainability science*, 12, 829-848 (2017).

de Miguel Ramos, C., & Laurenti, R.. Synergies and trade-offs among sustainable development goals: the case of Spain. *Sustainability*, 12(24), 10506 (2020).”

We have highlighted the goal-level synergies and trade-offs in Line 506-line 518 as follows:

“Based on the affiliation between indicator, target and goal⁴¹, the synergy/trade-off intensity was calculated as follows:

$$Intensity_{type} = Ratio_{type} \times ABS(R)_{type} \quad (1)$$

$$Ratio_{type} = \frac{EN_{type}}{TN_{type}} \quad (2)$$

$$ABS(R)_{type} = \frac{\sum_{i=1}^{EN_{type}} |R_{type}|}{EN_{type}} \quad (3)$$

where $Intensity_{type}$ was the synergy/trade-off intensity, $type$ refers to synergy/trade-off, $Ratio_{type}$ was the the ratio of the number of the selected indicator pairs out of the total number of all possible combinations among goals, $ABS(R)_{type}$ was the the absolute value of Spearman correlation coefficient R among goals. EN_{type} was the number of effective indicator pairs, TN_{type} was the total number of indicator pairs between goals, R_{type} was the Spearman correlation coefficient of the effective indicator pair. If we calculated the synergy and trade-off intensity directly at the goal level, we will ignore the fact that there were both synergies and trade-offs between different SDGs. ”

Issue 6: The explanation of synergies/trade-offs analysis at the provincial level (page 9) is a bit confusing. Why are indicator pairs presented as an aggregate of all 31 provinces (158,100 indicator pairs) instead of individually 5,100 pairs for each province? Then again on the explanation of networks (page 24), it says the networks were built separately (though at the goal level?).

Response: Revised as suggested.

Sorry for the confusing sentence. We have revised the explanation to make it clear as follows (lines 199-202):

“At the provincial level, 244-872 pairs showed synergies and 62-380 pairs showed trade-offs with the averaged $ABS(R)$ of 0.92-0.96 and 0.9-0.94, respectively (Bonferroni corrected $p < 0.05$ and $ABS(R) > 0.6$) (see supplementary Table 1 and the data file of “Synergies and trade-offs” for more details on each province).”

Issue 7: The explanation of network analysis methodology (page 24) is a bit vague. Maybe the authors can add more on the centralities and their denotations (including the choice of hub/authority centralities) and the overall process.

Response: Revised as suggested.

Thanks for the comments. We have added more details on the algorithm as follows (lines 527-548):

“We used the Hyperlink-Induced Topic Search (HITS) algorithm. This algorithm was proposed by Jon Kleinberg in 1999. It was originally used to sort web pages by importance and was later used for

network analysis. HITS adopts the principle of mutual reinforcement. It is based on the following two assumptions: (1) a high-quality authority node will be pointed to by many high-quality hub nodes; (2) a high-quality hub node will point to many high-quality authority nodes. Through multiple rounds of iterative calculations, each round of iterative calculations updates the two weights of each node until the weights are stable. The calculation formulas for the Authority (auth) value and Hub (hub) value of each node are as follows:

$$auth(p) = \sum_{q \in p_{to}} hub(q) \quad (1)$$

$$hub(p) = \sum_{q \in p_{from}} auth(q) \quad (2)$$

Among them, p is the target node, q is the other nodes, p_{to} represents the set of nodes pointed to node p from other nodes, and p_{from} represents the set of other nodes pointed from node p . The algorithm process is as follows:

- (1) Set the auth value and hub value of each node to 1;
- (2) Use formulas to calculate and update the auth value of each node;
- (3) Use the updated auth value to calculate and update the hub value of the node;
- (4) Normalize the auth value and hub value;
- (5) Repeat 2~4 until convergence or the stop iteration condition is reached.

For undirected matrices, the hub scores are the same as authority scores. The “hub_score” function from the “igraph” package was used to calculate the hub score of each network in our analysis. For further details of the algorithm, please refer to Kleinberg (1999).”

Issue 8: One additional observation: On page 3 (and in the abstract), the authors hold that understanding the synergies and trade-offs for determining the SDG priorities is essential “to rescue the SDGs from failing”. However, there are several other critical reasons – such as securing financing (particularly for developing countries) – that are equally (if not more) important for the effective implementation of the SDGs. Halfway through the implementation of the SDGs with no region or country of the world being on track in achieving the SDGs, it may be realistic not to suggest that one single approach will be effective in rescuing the SDGs.

Response: Revised as suggested.

Thanks for the comments. We realized the problem of the expression has revised the text as follows (line 65):

“One reason”

Part 2: Responses to reviewer 2

Reviewer #2 (Remarks to the Author):

The paper provides a provocatively titled analysis of the potential impacts of gender equity and climate on achieving the SDGs in China. It, moreover, offers a useful and technically sound analysis of the synergies and trade-offs between the SDGs at the local and national levels in China. However, the article has some significant problems in my view in its framing, presentation of results, and policy implications. To be considered for publication, major revisions are needed.

Response: Thanks for the comments. We have revised the framing, presentation of results and policy implications based on the specific comments below.

Issue 1: Framing: Consider a new title that emphasizes the local level interlinkages analysis and its implications in China. In my view, the paper does not really focus as much on gender and climate interactions as offering a local level intranational comparison of synergies and trade-offs in China. That is the data gap highlighted in the paper but it is not featured in the title.

Response: Revised as suggested.

Thanks for the comments. We have revised the title as follows (line 1-line 2):

“Intranational synergies and trade-offs reveal common and differentiated priorities of the Sustainable Development Goals in China”

Issue 2: Presenting Results: Consider a more focused and detailed explanation of what are the results for different regions; and what the intranational variation implies for a larger country like China. The current presentation is rather scatter shot with Tibet and Xinjiang having this kind of trade-off but not much discussion of the context and the reasons that such trade-offs arise. A similar claim could be made the synergies arguments that, in my view, lack the context and depth one might desire. To address this, I would suggest one or two case studies that go into greater depth on provincial level synergies and trade-offs.

Response: Revised as suggested.

Thanks for the comments. We have added the provincial level analysis on synergies and trade-offs with more discussion on the context and the reasons as follows (line 202- line 276):

“At the national level, we can assess the overall situation across the country, while at the provincial level, we can see the differences between different provinces within the country. Among goals, we found that SDG13 (Climate Action) and SDG5 (Gender Equality) had the lower hub scores in synergies on average (0.19 and 0.34) (Fig. 3(a), 3(c) and 3(e)) and higher in trade-offs (0.76 for both) for 19 provinces with a consistency of 19/31 as the national goals (Fig. 3(b), 3(d) and 3(f)). 14 of them had the highest trade-offs in SDG5 (Gender Equality) and 5 of them had the highest trade-offs in SDG13 (Climate Action) (see supplementary text of the results at the provincial level in SI for more details on the 19 provinces). The highest goal in trade-off differed among the other 12 provinces with a difference of 12/31, including SDG8 (Decent Work and Economic Growth) for Beijing and Chongqing, SDG10 (Reduced Inequalities) for Xinjiang, SDG3 (Good Health and Well-being) for Tibet, SDG9

(Industry, Innovation and Infrastructure) for Heilongjiang, SDG12 (Responsible Consumption and Production) for Jilin, SDG17 (Partnerships for the Goals) for Guangdong, Shanghai, and Qinghai, SDG16 (Peace, Justice and Strong Institutions) for Henan, and SDG2 (Zero Hunger) for Hubei and Tianjin (Fig. 3(b) and 3(d)). At the regional level, SDG13 (Combating Climate Change) and SDG5 (Gender Equality) dominated all the regions except for Northeast China, where SDG4 (Quality Education) had the highest trade-off (Fig. 4(e) and 4(f)). The trade-off in SDG5 (Gender Equality) in southern regions were also higher than the northern regions (Fig. 4(e)). The trade-off in SDG13 (Combating Climate Change) presented an opposite pattern between north and south, i.e. it increased from east to west in the north, while it decreased in the south (Fig. 4 (f)).

We found that most of the SDGs had the higher scores in synergies than trade-offs. Among them, SDG1 (No Poverty) and SDG6 (Clean Water and Sanitation) showed the high scores in synergies (0.98 and 0.97) (Fig. 3(a), 3(c) and 3(e)) and lower in trade-offs (0.35 and 0.42) for 24 provinces with a consistency of 24/31 as the national ones. Among the 24 provinces, 14 of them had the highest synergy in SDG1 (No Poverty) and 10 of them had the highest synergy in SDG6 (Clean Water and Sanitation) (Fig. 3(b), 3(d) and 3(f), see supplementary text of the results at the provincial level in SI for more details on the 24 provinces). The highest goal in synergy differed among the other 7 provinces, including SDG2 for Henan, SDG11 (Sustainable cities and Communities) for Inner Mongolia and Jilin, SDG7 (Affordable and Clean Energy) for Jiangxi, SDG4 (Quality Education) for Sichuan, Tibet and Xinjiang (Fig. 3(a) and 3(c), see supplementary text of the results at the provincial level in SI for more details on the 7 provinces). At the regional level, we also found that SDG1 (No Poverty) and SDG6 (Clean Water and Sanitation) dominated all the 6 regions in synergy (Fig. 4(a)). These findings clearly reflect the existence of common but different priorities for SDGs in China at the national and provincial levels.

Each province or region presents different synergy and trade-off priorities due to its own geographical location, resource endowment, climatic conditions, topography, historical development foundation and other factors. The varied SDG priorities reflect that based on ensuring a coordinated national response, China should pay more attention to the actual situation of each province, so that provincial governments can formulate more targeted policies in line with regional SDGs priorities towards high-quality sustainable development.

Take Tibet with the lowest GDP in China as an example to discuss the provincial level trade-offs. Restricted by natural, geographical, climatic and historical factors, the overall development of medical and health services in Tibet was still relatively lagging behind, with poor conditions and low levels. The total amount of medical and health resources was insufficient and unevenly distributed, and medical service capabilities are weak. From the perspective of institutional mechanisms, problems were still prominent, such as extensive management methods, insufficient strict implementation of the medical system, and insufficient procurement of urgently needed drugs. From the perspective of behavioral concept, the public's awareness of health was not strong, and the problem of diseases caused by unhealthy lifestyles was obvious⁵¹. Looking at the disease spectrum, the incidence rates of AIDS, tuberculosis, and hepatitis B per 100,000 people increased from 0.02, 75.07, and 52.62 in 2002 to 1.34, 150.13, and 107.29 in 2020, respectively⁵². The combined constraints of the natural, geographical, climatic and historical factors make SDG3 (Good Health and Well-being) in Tibet have the highest trade-offs.

Take Guangdong as an example to discuss the provincial level synergies. Guangdong as the forefront of reform and opening up, had the highest GDP in China. However, there was still a large amount of poverty in the mountainous areas in the north and the underdeveloped areas in the west due to remote geographical location, numerous mountainous areas, insufficient transportation infrastructure, and single industrial structure. To this end, Guangdong government integrated the targeted poverty alleviation campaign into the overall economic and social development plan for overall planning, fully leveraged the synergy between SDG1 (No Poverty) and different SDGs, established a poverty reduction governance pattern that coordinates and promotes coordination and mutual promotion, and focused on stimulating the endogenous motivation of the poor to get rid of poverty. Through precise strategies, policies were implemented accurately targeted to villages, households and people. Through promoting the combination of the market and the government leverages, different programs were implemented, including special poverty alleviation, industry poverty alleviation, social poverty alleviation⁵³. From 2013 to 2022, a total of 2.502 million poor people in the province were lifted out of poverty, and per capita disposable income increased by more than 2.6 times⁵⁴. Through targeted poverty alleviation and poverty alleviation efforts, Guangdong was at the forefront of the country, making SDG1 (No Poverty) the highest goal in synergies. In the future, Guangdong needed to continue to leverage the synergistic advantages of SDG1 (No Poverty) and gradually narrowed the income gap in the process of further getting rid of relative poverty and realizing the rural revitalization strategy.

Adhere to the people-centered development philosophy and strive to solve the difficulties in building a healthy Tibet, <https://www.12371.cn/2019/07/19/ARTI1563527888637660.shtml>, accessed on March, 2023 [in Chinese].

China Statistics Press, “China Health and Family Planning Statistical Yearbook” (National Bureau of Statistics of the People's Republic of China, 2001-2021) [in Chinese].

Making good use of the key strategy of reform and opening up, Guangdong has contributed valuable experience to targeted poverty reduction. <https://baijiahao.baidu.com/s?id=1703415407980845546&wfr=spider&for=pc>, accessed on Nov. 6, 2023 [In Chinese].

In the past ten years, a total of 2.502 million poor people in the province have been lifted out of poverty and rural areas have been comprehensively revitalized. Guangdong has embarked on a new journey. https://www.gd.gov.cn/zwgk/zdlyxxgkzl/fpgzxx/content/post_4018772.html, accessed on Nov. 6, 2023 [In Chinese].”

Issue 3: Policy Implications: Here again, I found the discussion of policy thin and weak. I would suggest a greater emphasis on underlining how large countries such as China can take advantage of fiscal reallocation policies and administrative staffing mechanisms to limit trade-offs and leverage synergies. For example, there is a long history of China using fiscal policy to move resources from more progressive to regions that lagging in socioeconomic development. Similarly, there is also the potential—more so that in a modern democracy—to move local leaders to a region where there is a need for innovative thinking on the links between climate and gender. Please check the China studies literature to offer some suggestions that consider these possibilities. My fear, otherwise, is that the article is based chiefly on a modelling analysis that lacks the empirical richness needed to make it really useful for China and other countries seeking to balance competing priorities.

Response: Revised as suggested.

Thanks for the comments. We have added the discussion related to fiscal reallocation policies and administrative staffing mechanisms as follows (line 345- line 373):

“Large countries such as China can take advantage of fiscal reallocation policies and administrative staffing mechanisms to limit trade-offs and leverage synergies. There was a long history of China using fiscal policy to move resources from more progressive to regions that lagging in socioeconomic development. Through the fiscal transfer payment policy, the central government transferred the fiscal surplus of high-GDP provinces such as Guangdong, Shanghai, and Beijing to Sichuan, Hunan, Hubei, Yunnan and other provinces to ensure education (SDG4), medical care (SDG3), balance the development of public services (SDG1) and narrow the gap on development among regions (SDG10) (Central transfer payments exceeded 10 trillion yuan for the first time, how much money was allocated to each province?, accessed on Oct. 26, 2023). Our results indicate that the health goal in Tibet (SDG3) and the social inequality in Xinjiang (SDG10) has the greatest trade-offs, and are areas where traditional fiscal transfer payments need to be focused. The central government can also consider expanding the service scope of fiscal transfer payments. For example, in the provinces or regions where SDG13 (Combating Climate Change) dominates, fiscal transfer payment support can be considered to increase for Inner Mongolia, Jiangxi, Ningxia, Shaanxi, and Shanxi to address the challenge. In terms of SDG5 (Gender Equality), the increased support can occur in Anhui, Gansu, Guangxi, Guizhou, Hainan, Hebei, Hunan, Sichuan, and Yunnan, which have the most prominent trade-offs in SDG5 (Gender Equality), and are fiscally concerned with financial income less than expenditure. While Fujian, Liaoning, Shandong, Zhejiang, and Jiangsu can increase their support for SDG5 based on their own finances as they all have financial surplus.

Similarly, there is also the potential—more so that in a modern democracy—to move local leaders to a region where there is a need for innovative thinking on the links between climate and gender. Based on the performance highlights and comprehensive qualities of provincial leaders in the past, leaders can be moved to other provinces to promote learning from the experience of advanced provinces in dealing with SDGs in trade-offs and synergies (Li et al., 2022). For example, Tianjin has high synergies in SDG5 (Gender Equality), which is worthy of reference by all other provinces. Tibet with the highest trade-offs in SDG3 (Good Health and Well-being) can learn from the experience of Yunnan, which has greater synergy benefits in the health goal. Tianjin and Hubei having the highest trade-offs in SDG2 (Zero Hunger) can learn from Henan which obtain the highest synergies. Heilongjiang with the highest trade-offs in SDG9 can learn from the experience of Anhui and Qinghai with high synergies in this goal.

Central transfer payments exceeded 10 trillion yuan for the first time, how much money was allocated to each province? <https://baijiahao.baidu.com/s?id=1763142399541846936&wfr=spider&for=pc>, accessed on Oct. 26, 2023 [In Chinese].

Li, Y., Shao, X., Tao, Z., & Yuan, H.. How local leaders matter: Inter-provincial leadership transfers and land transactions in China. *Journal of Comparative Economics*, 50(1), 196-220 (2022).”

Part 3: Responses to reviewer 3

Reviewer #3 (Remarks to the Author):

Review Report: Climate action and gender equality matter most for China's sustainable development

While the paper offers valuable insights into the complex interactions among SDGs in China, there are areas in each section that can be improved to enhance the overall impact and utility of the research. Addressing these areas for improvement will contribute to a more robust and comprehensive research paper. Overall, the abstract and introduction provide a good starting point for the paper but can be improved by including more specific findings and implications and enhancing their engagement with the reader. Additionally, providing more context and citations could further support the research's significance.

The Results section provides valuable quantitative insights into the priority of SDGs at the national level in China. To enhance this section, the authors should provide more detailed explanations of metrics, offer a deeper interpretation of results, discuss data quality, and make explicit connections to the research questions and implications for policy and practice.

The "Discussion and Policy Implication" section effectively outlines the policy implications of the research findings, emphasising the importance of addressing climate action and gender equality while harnessing the positive impacts of poverty alleviation and clean water initiatives. To enhance this section, the authors should provide a deeper analysis, explore cross-boundary interactions, connect findings to the global context, offer specific policy recommendations, and conclude with a concise summary of key takeaways.

Response: Thanks for the comments. We have made revisions based the detailed comments for each section.

Some specific points in these sections are slated below.

1. The paper effectively communicates its research objectives and the identified research gap in the abstract section. It excels in clarity, making the central focus of the research readily understandable. However, there are areas where improvement is needed, such as the absence of specific findings and quantitative data, the lack of clear implications, the need for specific keywords to enhance discoverability, and the potential for a bit more detail to convey key insights effectively. Kindly consider some of these points:

Issue 1: While it mentions identifying climate change (SDG13) and gender equality (SDG5) as key hurdles, it lacks specific findings and quantitative data to make the abstract more informative and enticing to potential readers.

Response: Revised as suggested.

Thanks for the comments. We have added the quantitative information in the abstract as follows (line 45- line 55):

“Our results reveal that 19 provinces show the highest trade-offs in SDG13 (Combating Climate Change) or SDG5 (Gender Equality) consistent with the national level, with other 12 provinces showing different goals as top trade-offs, including SDG2 (Zero Hunger), SDG3 (Good Health and Well-being), SDG8 (Decent Work and Economic Growth), SDG9 (Industry, Innovation and Infrastructure), SDG10 (Reduced Inequalities), SDG 12 (Responsible Consumption and Production), SDG16 (Peace, Justice and Strong Institutions) and SDG17 (Partnerships for the Goals). 24 provinces show the highest synergies in SDG1 (No Poverty) or SDG6 (Clean Water and Sanitation) consistent with the national level, with the remaining 7 provinces showing different top synergies goals, including SDG2 (Zero Hunger), SDG4 (Quality Education), SDG7 (Affordable and Clean Energy) and SDG11 (Sustainable Cities and Communities).”

Issue 2: The abstract does not clearly indicate why these findings are important or what implications they might have for China's sustainable development.

Response: Revised as suggested.

Thanks for the comments. We have revised the text to make it clear on the implications for China's sustainable development as follows (line 55- line 58):

“These common but differentiated SDG priorities reflect that to ensure a coordinated national response, China should pay more attention to the actual situation of each province, so that provincial governments can formulate more targeted policies in line with provincial SDG priorities towards accelerating sustainable development.”

Issue 3: Including specific keywords related to climate change, gender equality, and SDGs could enhance the discoverability of the paper.

Response: Revised as suggested.

Thanks for the comments. We have added the keywords in the revised version (line 59) as follows:

“Keywords: SDGs, climate change, gender equality, interactions, sustainable development”

Issue 4: While being concise is important, the abstract could benefit from more detail, particularly in summarising key findings or insights.

Response: Revised as suggested.

Thanks for the comments. We have integrated this comment with issue 1 and issue 2 to add the summary of key findings as follows (line 45- line 58):

“Our results reveal that 19 provinces show the highest trade-offs in SDG13 (Combating Climate Change) or SDG5 (Gender Equality) consistent with the national level, with other 12 provinces showing different goals as top trade-offs, including SDG2 (Zero Hunger), SDG3 (Good Health and

Well-being), SDG8 (Decent Work and Economic Growth), SDG9 (Industry, Innovation and Infrastructure), SDG10 (Reduced Inequalities), SDG 12 (Responsible Consumption and Production), SDG16 (Peace, Justice and Strong Institutions) and SDG17 (Partnerships for the Goals). 24 provinces show the highest synergies in SDG1 (No Poverty) or SDG6 (Clean Water and Sanitation) consistent with the national level, with the remaining 7 provinces showing different top synergies goals, including SDG2 (Zero Hunger), SDG4 (Quality Education), SDG7 (Affordable and Clean Energy) and SDG11 (Sustainable Cities and Communities). These common but differentiated SDG priorities reflect that to ensure a coordinated national response, China should pay more attention to the actual situation of each province, so that provincial governments can formulate more targeted policies in line with provincial SDG priorities towards accelerating sustainable development.”

2. The introduction provides a solid foundation for the paper, setting the stage by introducing the 2030 Agenda and justifying China's selection as a case study. It also clearly identifies the research gap related to SDG interactions. However, it could benefit from more explicit relevance of gender and climate goals to China's sustainable development, a smoother transition to the methodology section, a more engaging introduction, and more comprehensive citations to support its arguments. Some points to consider:

Issue 5: Explicit Relevance of Gender and Climate: While the introduction mentions that combating climate change (SDG13) and improving gender equality (SDG5) are key hurdles, it could be more explicit about why these specific goals are relevant to China's sustainable development.

Response: Revised as suggested.

Thanks for the comments. We have added the relevance of Gender and Climate (line 91-98) as follows: “For example, climate change has exerted persistent impacts on China’s ecological environment and socioeconomic development and brought serious threats to its food, water, ecology, energy, and urban operation security, as well as people’s safety and property (Fuso et al., 2019). China’s carbon emissions have significantly increased by around 10 times over the past 50 years (Guan et al., 2021).

Fuso Nerini, F., Sovacool, B., Hughes, N., Cozzi, L., Cosgrave, E., Howells, M., ... & Milligan, B. Connecting climate action with other Sustainable Development Goals. *Nat. Sustain.* 2(8), 674-680 (2019).

Guan, Y., Shan, Y., Huang, Q., Chen, H., Wang, D., & Hubacek, K. Assessment to China's Recent Emission Pattern Shifts. *Earths Future* 9 (11), e2021EF002241 (2021).

Gender equality plays an important role in improving productivity and reducing gender discrimination and violence to promote economic development and social progress (Achieve Gender Equality to Deliver the SDGs, accessed on March 22, 2023). However, the gender gap in labour force participation between men and women rose from 9% to almost 15% between the 1990s and 2020 (Liang et al., 2021).

Achieve Gender Equality to Deliver the SDGs, sdg.iisd.org/commentary/policy-briefs/achieve-gender-equality-to-deliver-the-sdgs/, accessed on March 22, 2023.

Liang, D., Lu, X., Zhuang, M., Shi, G., Hu, C., Wang, S., & Hao, J.. China's greenhouse gas emissions for cropping systems from 1978–2016. *Scientific Data*, 8(1), 171 (2021).”

Issue 6: Transition to Methodology: The transition from the research gap to the methodology section could be smoother. It would be helpful to briefly introduce how the study aims to address the research gap before diving into the methodology.

Response: Revised as suggested.

Thanks for the comments. We have added a brief introduction on how the study aims to address the research gap (104-105) as follows:

“we aim to address the research gaps by analyzing the synergies and trade-offs among the 17 SDGs in China at the national, provincial and regional levels.”

Issue 7: Engagement of the Reader: The introduction could be more engaging by using more vivid language or a compelling statistic to draw the reader's attention to the importance of the research.

Response: Revised as suggested.

Thanks for the comments. We have added statistics (line 93-line 94 and line 96-line 98) as follows:

“China's carbon emissions have significantly increased by around 10 times over the past 50 years (Guan et al., 2021).

Guan, Y., Shan, Y., Huang, Q., Chen, H., Wang, D., & Hubacek, K. Assessment to China's Recent Emission Pattern Shifts. *Earths Future* 9 (11), e2021EF002241 (2021).

The gender gap in labour force participation between men and women rose from 9% to almost 15% between the 1990s and 2020 (Liang et al., 2021).

Liang, D., Lu, X., Zhuang, M., Shi, G., Hu, C., Wang, S., & Hao, J.. China's greenhouse gas emissions for cropping systems from 1978–2016. *Scientific Data*, 8(1), 171 (2021).”

Issue 8: Citations: It could benefit from citing specific studies or reports highlighting the challenges and disparities in SDG progress within China. This would strengthen the foundation of the research.

Response: Revised as suggested.

Thanks for the comments. We have added citations to highlight the challenges and disparities in SDG progress within China (line 98-100, line 83-line84) as follows:

“Further, studies highlighted the challenges and disparities in SDG progress within China, suggesting that the uneven progress among the 17 SDGs at the sub-national level is a significant challenge for China's sustainable development (Kuhn et al., 2018; Lu et al., 2019).

Lu, Y., Zhang, Y., Cao, X., Wang, C., Wang, Y., Zhang, M., ... & Zhang, Z.. Forty years of reform and opening up: China's progress toward a sustainable path. *Science advances*, 5(8), eaau9413 (2019).

Kuhn, B. M. (2018). China's commitment to the sustainable development goals: An analysis of push and pull factors and implementation challenges. *Chinese Political Science Review*, 3(4), 359-388.

Understanding SDGs interactions at these sub-national levels is essential since it is where the SDGs are implemented (Wang et al., 2020).

Wang, Y., Lu, Y., He, G., Wang, C., Yuan, J., & Cao, X.. Spatial variability of sustainable development goals in China: A provincial level evaluation. *Environmental Development*, 35, 100483 (2020).”

3. Results: The results section offers valuable quantitative insights into the priority of SDGs at the national level in China. Strengths include quantitative rigour, effective visualisation, clear identification of key goals, and a comparison between synergy and trade-off networks. Areas for improvement include a more detailed explanation of metrics, a deeper interpretation of results, a discussion of data quality and reliability, references to supplementary information, and a stronger connection to research questions. The section analyses the priority of Sustainable Development Goals (SDGs) at the national level in China based on synergy and trade-off networks. Some points to consider:

Issue 9: Detailed Explanation of Metrics: While the section mentions ABS(R) and the Ratio, it could benefit from a more detailed explanation of these metrics and their significance. Readers may not be familiar with these measures, and providing a brief description or formula would enhance clarity.

Response: Revised as suggested.

Thanks for the comments. We have added a more detailed explanation and the formula of ABS(R) and the Ratio (line 506-518) as follows:

“Based on the affiliation between indicator, target and goal⁴¹, the synergy/trade-off intensity was calculated as follows:

$$Intensity_{type} = Ratio_{type} \times ABS(R)_{type} \quad (1)$$

$$Ratio_{type} = \frac{EN_{type}}{TN_{type}} \quad (2)$$

$$ABS(R)_{type} = \frac{\sum_{i=1}^{EN_{type}} |R_{type}|}{EN_{type}} \quad (3)$$

Where $Intensity_{type}$ was the synergy/trade-off intensity, $type$ refers to synergy/trade-off, $Ratio_{type}$ was the the ratio of the number of the selected indicator pairs out of the total number of all possible combinations among goals, $ABS(R)_{type}$ was the the absolute value of Spearman correlation coefficient R among goals. EN_{type} was the number of effective indicator pairs, TN_{type} was the total number of indicator pairs between goals, R_{type} was the Spearman correlation coefficient of the effective indicator pair. If we calculated the synergy and trade-off intensity directly at the goal level, we will ignore the fact that there were both synergies and trade-offs between different SDGs.”

Issue 10: Interpretation of Results: The section could improve by providing more interpretation. For example, why do SDG13 and SDG5 have the highest hub scores in the trade-off network? What are the practical implications of these findings for China's sustainable development? A deeper analysis of the "why" behind these results would strengthen the section.

Response: Revised as suggested.

Thanks for the comments. We have added more interpretation of the results on the reasons and the practical implications of these findings as follows (line 134-line 159):

“China became the world's largest emitter of carbon dioxide (CO₂) in 2006³⁰. China slipped from 63rd position in 2006 to 106th in the global gender gap rankings among 153 countries in 2019³⁹. These brought serious trade-offs in SDG13 (Climate Action) and SDG5 (Gender Equality) along with the rapid economic development. Given combatting climate change can reinforce all 17 SDGs²⁹. Gender equality is an enabler and accelerator for all the SDGs³¹, the most important is that China overall needs to take decisive actions to mitigate the negative impact from SDG13 (Climate Action) and SDG5 (Gender Equality). These findings address explicitly the common priorities of the SDGs in trade-off (SDG13 and SDG5) and synergy (SDG1 and SDG6) among the different spatial levels in China.

China's trade-offs in SDG13 (Climate Action) and SDG5 (Gender Equality) are at a high level, and it needs to increase efforts at the national level for top-level design. For SDG13 (Climate Action), China needs to strengthen the top-level design of carbon peak and carbon neutral, propose a systematic, all-round plan for every sector, lead all-round green transformation to combat climate change. On one side, it requires reductions in high carbon emissions from economic development (SDG8)³⁰, especially the industry (SDG9) and traditional energy sectors (SDG7)³⁰, and from agriculture (SDG2)³²; on the other side, it requires reinforcing the carbon sink in China's terrestrial ecosystems (SDG15)³³. Besides, it also needs reducing the threat to water supplies and sanitation services (SDG6)³⁴⁻³⁵, quality education (SDG4)³⁹⁻⁴⁰, and human health (SDG3)⁴¹⁻⁴². For SDG5 (Gender Equality), the central government should take the lead to strengthen the top-level design and launch a package of much stronger plans and measures in every sector in the short and long terms to promote the all-round development of women and girl. In particular, China should reduce gender gap in women's employment rate and wage (SDG8 and SDG9)⁴³, participation in decision-making (SDG5)⁴⁴, healthcare (SDG3)⁴⁵, rural and secondary education (SDG4)^{44, 46}, poverty reduction (SDG1)⁴⁷, rights protection in agriculture, forestry and animal husbandry (SDG2 and SDG15)⁴⁸, water and sanitation (SDG6)⁴⁹, and building partnership (SDG17)⁵⁰.”

Issue 11: Data Quality and Reliability: Given the extensive dataset used for analysis, it would be helpful to briefly discuss the quality and reliability of the data. Were there any challenges or limitations in data collection that could affect the results?

Response: Revised as suggested.

Thanks for the comments. We have added the discussion on the quality and reliability of the data as follows (line 160- line 170):

“We chose the SDG indicators with the most available data at the national and sub-national levels simultaneously in our study to ensure the reliability of the results. The data came from a variety of official statistical yearbooks. Each type of data was collected by the corresponding official national ministries and provincial counterparts and was the most authoritative data currently available. Faced with such large-scale data collection, there were indeed varying degrees of data missing problems

across the country and in different provinces due to differences in data collection capabilities, local conditions, personnel, budgets, etc. At the national scale, none of the indicators show no data and the data integrity is overall good. The ratio of indicators covering more than 15 years accounted for over 94%. The indicator of 9.c.1 with least data still covered 7 years. The lack of data in some years was mainly because the indicators were developed later and had not been collected by the official statistics before.”

Issue 12: References to Supplementary Information: The section references supplementary information (SI) for more details. However, it could be improved by briefly summarising the key findings or insights from the supplementary information, making it more accessible to readers.

Response: Revised as suggested.

Thanks for the comments. We have moved most of the content in the supplementary information to the main text as they are all closely related to the reviewer’s comments. For some references to Supplementary Information, we have added a brief summary as follows (line 122-123, line 132-133) :

“Please see the spreadsheet named “National” in the supplementary file of “synergies and trade-offs” for more details on the different category in Fig. 1(a).

Please see supplementary text in SI for more details on the priorities of SDGs at the national level.”

Issue 13: Connection to Research Questions: While the section discusses findings related to SDG priorities, it could be more explicit in connecting these findings to the research questions posed in the introduction. How do these findings address common priorities among different spatial levels in China?

Response: Revised as suggested.

Thanks for the comments. We have revised the text to explicitly connecting the findings to address the common priorities among various levels in China as follows (line 140-line 142):

“These findings address explicitly the common priorities of the SDGs in trade-off (SDG13 and SDG5) and synergy (SDG1 and SDG6) among the different spatial levels in China.”

4. Section - Similarity and Differences of SDG Priority among Provinces: This section analyses SDG priorities at the provincial level in China, presenting quantitative data and comparisons across goals. Strengths include quantitative rigour, effective visualisation, regional analysis, and comparisons. However, it could benefit from a deeper interpretation of results, a stronger connection to research questions, a discussion of data quality, integration with previous findings, and an exploration of policy implications for regional development. The section provides valuable quantitative insights into SDG priorities in China’s provincial level, focusing on synergy and trade-off networks. To enhance this section, the authors should provide a deeper interpretation of results, connect findings to research questions, discuss data quality, integrate findings with previous results, and explore policy implications for regional development.

Issue 14: Interpretation of Results: While the section presents the data and identifies differences in SDG priorities, it could benefit from a deeper interpretation of the results. Why do certain provinces or regions prioritise specific SDGs over others? What are the implications of these variations for policy and development strategies?

Response: Revised as suggested.

Thanks for the comments. We have added more deeper interpretation of the results on the reasons and the implications as follows (line 202- line 276):

“At the national level, we can assess the overall situation across the country, while at the provincial level, we can see the differences between different provinces within the country. Among goals, we found that SDG13 (Climate Action) and SDG5 (Gender Equality) had the lower hub scores in synergies on average (0.19 and 0.34) (Fig. 3(a), 3(c) and 3(e)) and higher in trade-offs (0.76 for both) for 19 provinces with a consistency of 19/31 as the national goals (Fig. 3(b), 3(d) and 3(f)). 14 of them had the highest trade-offs in SDG5 (Gender Equality) and 5 of them had the highest trade-offs in SDG13 (Climate Action) (see supplementary text of the results at the provincial level in SI for more details on the 19 provinces). The highest goal in trade-off differed among the other 12 provinces with a difference of 12/31, including SDG8 (Decent Work and Economic Growth) for Beijing and Chongqing, SDG10 (Reduced Inequalities) for Xinjiang, SDG3 (Good Health and Well-being) for Tibet, SDG9 (Industry, Innovation and Infrastructure) for Heilongjiang, SDG12 (Responsible Consumption and Production) for Jilin, SDG17 (Partnerships for the Goals) for Guangdong, Shanghai, and Qinghai, SDG16 (Peace, Justice and Strong Institutions) for Henan, and SDG2 (Zero Hunger) for Hubei and Tianjin (Fig. 3(b) and 3(d)). At the regional level, SDG13 (Combating Climate Change) and SDG5 (Gender Equality) dominated all the regions except for Northeast China, where SDG4 (Quality Education) had the highest trade-off (Fig. 4(e) and 4(f)). The trade-off in SDG5 (Gender Equality) in southern regions were also higher than the northern regions (Fig. 4(e)). The trade-off in SDG13 (Combating Climate Change) presented an opposite pattern between north and south, i.e. it increased from east to west in the north, while it decreased in the south (Fig. 4 (f)).

We found that most of the SDGs had the higher scores in synergies than trade-offs. Among them, SDG1 (No Poverty) and SDG6 (Clean Water and Sanitation) showed the high scores in synergies (0.98 and 0.97) (Fig. 3(a), 3(c) and 3(e)) and lower in trade-offs (0.35 and 0.42) for 24 provinces with a consistency of 24/31 as the national ones. Among the 24 provinces, 14 of them had the highest synergy in SDG1 (No Poverty) and 10 of them had the highest synergy in SDG6 (Clean Water and Sanitation) (Fig. 3(b), 3(d) and 3(f), see supplementary text of the results at the provincial level in SI for more details on the 24 provinces). The highest goal in synergy differed among the other 7 provinces, including SDG2 for Henan, SDG11 (Sustainable cities and Communities) for Inner Mongolia and Jilin, SDG7 (Affordable and Clean Energy) for Jiangxi, SDG4 (Quality Education) for Sichuan, Tibet and Xinjiang (Fig. 3(a) and 3(c), see supplementary text of the results at the provincial level in SI for more details on the 7 provinces). At the regional level, we also found that SDG1 (No Poverty) and SDG6 (Clean Water and Sanitation) dominated all the 6 regions in synergy (Fig. 4(a)). These findings clearly reflect the existence of common but different priorities for SDGs in China at the national and provincial levels.

Each province or region presents different synergy and trade-off priorities due to its own geographical location, resource endowment, climatic conditions, topography, historical development foundation and other factors. The varied SDG priorities reflect that based on ensuring a coordinated

national response, China should pay more attention to the actual situation of each province, so that provincial governments can formulate more targeted policies in line with regional SDGs priorities towards high-quality sustainable development.

Take Tibet with the lowest GDP in China as an example to discuss the provincial level trade-offs. Restricted by natural, geographical, climatic and historical factors, the overall development of medical and health services in Tibet was still relatively lagging behind, with poor conditions and low levels. The total amount of medical and health resources was insufficient and unevenly distributed, and medical service capabilities are weak. From the perspective of institutional mechanisms, problems were still prominent, such as extensive management methods, insufficient strict implementation of the medical system, and insufficient procurement of urgently needed drugs. From the perspective of behavioral concept, the public's awareness of health was not strong, and the problem of diseases caused by unhealthy lifestyles was obvious⁵¹. Looking at the disease spectrum, the incidence rates of AIDS, tuberculosis, and hepatitis B per 100,000 people increased from 0.02, 75.07, and 52.62 in 2002 to 1.34, 150.13, and 107.29 in 2020, respectively⁵². The combined constraints of the natural, geographical, climatic and historical factors make SDG3 (Good Health and Well-being) in Tibet have the highest trade-offs.

Take Guangdong as an example to discuss the provincial level synergies. Guangdong as the forefront of reform and opening up, had the highest GDP in China. However, there was still a large amount of poverty in the mountainous areas in the north and the underdeveloped areas in the west due to remote geographical location, numerous mountainous areas, insufficient transportation infrastructure, and single industrial structure. To this end, Guangdong government integrated the targeted poverty alleviation campaign into the overall economic and social development plan for overall planning, fully leveraged the synergy between SDG1 (No Poverty) and different SDGs, established a poverty reduction governance pattern that coordinates and promotes coordination and mutual promotion, and focused on stimulating the endogenous motivation of the poor to get rid of poverty. Through precise strategies, policies were implemented accurately targeted to villages, households and people. Through promoting the combination of the market and the government leverages, different programs were implemented, including special poverty alleviation, industry poverty alleviation, social poverty alleviation⁵³. From 2013 to 2022, a total of 2.502 million poor people in the province were lifted out of poverty, and per capita disposable income increased by more than 2.6 times⁵⁴. Through targeted poverty alleviation and poverty alleviation efforts, Guangdong was at the forefront of the country, making SDG1 (No Poverty) the highest goal in synergies. In the future, Guangdong needed to continue to leverage the synergistic advantages of SDG1 (No Poverty) and gradually narrowed the income gap in the process of further getting rid of relative poverty and realizing the rural revitalization strategy.

Adhere to the people-centered development philosophy and strive to solve the difficulties in building a healthy Tibet, <https://www.12371.cn/2019/07/19/ARTI1563527888637660.shtml>, accessed on March, 2023 [in Chinese].

China Statistics Press, “China Health and Family Planning Statistical Yearbook” (National Bureau of Statistics of the People's Republic of China, 2001-2021) [in Chinese].

Making good use of the key strategy of reform and opening up, Guangdong has contributed valuable experience to targeted poverty reduction.

<https://baijiahao.baidu.com/s?id=1703415407980845546&wfr=spider&for=pc>, accessed on Nov. 6, 2023 [In Chinese].

In the past ten years, a total of 2.502 million poor people in the province have been lifted out of poverty and rural areas have been comprehensively revitalized. Guangdong has embarked on a new journey. https://www.gd.gov.cn/zwgk/zdlyxxgkzl/fpgzxx/content/post_4018772.html, accessed on Nov. 6, 2023 [In Chinese].”

Issue 15: Relevance to Research Questions: The section should explicitly connect the findings to the research questions in the introduction. How do these provincial variations in SDG priorities relate to the overall research objectives of understanding SDG interactions at different spatial levels?

Response: Revised as suggested.

Thanks for the comments. We have revised the text to explicitly connect the findings to understanding SDG interactions at various spatial levels as follows (line 277- line 280):

“The differences in the priorities of SDGs among China's 31 provinces provide the results of the priorities of different SDGs at the provincial level in terms of synergies and trade-offs for understanding the interaction of SDGs at different spatial levels in China. It is also the basis for obtaining the priority of SDGs at the regional level.”

Issue 16: Data Quality: Similar to the earlier part of the Results section, it would be beneficial to briefly discuss the quality and reliability of the data used for provincial-level analysis. This can help readers understand the potential limitations of the findings.

Response: Revised as suggested.

Thanks for the comments. We have added the discussion on the quality and reliability of the data as follows (line 290- line 300):

“At the provincial level, 30 provinces ha the number of indicators between 98 and 102. For Tibet with remote location and difficult conditions, 94 indicators were collected to its maximum extent. After massive efforts on data collection, overall we compiled a set of 102 SDG indicators (118 original indicator and 102 after calculation), 81 targets, and 17 goals, at the provincial level on a yearly basis, which was greater than the number of indicators in the 2022 SDG Index and Dashboards Report (which used 88 indicators to assess China's SDGs performances at the national level)⁵⁹ and one previous study (which used 88 indicators, 71 targets, and 16 goals to calculate the score of 16 goals in China and analyze the SDGs interactions between 3 general categories using the score at the goal level)⁶⁰. Our understanding of the SDGs interaction networks in China at provincial, regional and national levels will evolve as more data become available.”

Issue 17: Integration with Previous Findings: Integrating the provincial-level findings with the national-level findings discussed earlier in the Results section is essential. Are there consistencies or differences between the two levels of analysis? How do these findings contribute to understanding China's SDG priorities?

Response: Revised as suggested.

Thanks for the comments. We have added the consistencies and differences between the provincial and national levels as follows (line 202-line 216, line 223- line 233, line 235-237):

“At the national level, we can assess the overall situation across the country, while at the provincial level, we can see the differences between different provinces within the country. We found that most of the SDGs had the higher scores in synergies than trade-offs. Among them, SDG1 (No Poverty) and SDG6 (Clean Water and Sanitation) showed the high scores in synergies (0.98 and 0.97) (Fig. 3(a), 3(c) and 3(e)) and lower in trade-offs (0.35 and 0.42) for 24 provinces with a consistency of 24/31 as the national ones. Among the 24 provinces, 14 of them had the highest synergy in SDG1 (No Poverty) and 10 of them had the highest synergy in SDG6 (Clean Water and Sanitation) (Fig. 3(b), 3(d) and 3(f), see supplementary text of the results at the provincial level in SI for more details on the 24 provinces). The highest goal in synergy differed among the other 7 provinces, including SDG2 for Henan, SDG11 (Sustainable cities and Communities) for Inner Mongolia and Jilin, SDG7 (Affordable and Clean Energy) for Jiangxi, SDG4 (Quality Education) for Sichuan, Tibet and Xinjiang (Fig. 3(a) and 3(c), see supplementary text of the results at the provincial level in SI for more details on the 7 provinces).

Among goals, we found that SDG13 (Climate Action) and SDG5 (Gender Equality) had the lower hub scores in synergies on average (0.19 and 0.34) (Fig. 3(a), 3(c) and 3(e)) and higher in trade-offs (0.76 for both) for 19 provinces with a consistency of 19/31 as the national goals (Fig. 3(b), 3(d) and 3(f)). 14 of them had the highest trade-offs in SDG5 (Gender Equality) and 5 of them had the highest trade-offs in SDG13 (Climate Action) (see supplementary text of the results at the provincial level in SI for more details on the 19 provinces). The highest goal in trade-off differed among the other 12 provinces with a difference of 12/31, including SDG8 (Decent Work and Economic Growth) for Beijing and Chongqing, SDG10 (Reduced Inequalities) for Xinjiang, SDG3 (Good Health and Well-being) for Tibet, SDG9 (Industry, Innovation and Infrastructure) for Heilongjiang, SDG12 (Responsible Consumption and Production) for Jilin, SDG17 (Partnerships for the Goals) for Guangdong, Shanghai, and Qinghai, SDG16 (Peace, Justice and Strong Institutions) for Henan, and SDG2 (Zero Hunger) for Hubei and Tianjin (Fig. 3(b) and 3(d)).

These findings clearly reflect the existence of common but different priorities for SDGs in China at the national and provincial levels.”

Issue 18: Policy Implications: The section lacks discussion of the practical implications of the provincial-level findings for policymakers and development practitioners. A brief discussion of how these insights can inform regional development strategies or SDG implementation plans would be valuable.

Response: Revised as suggested.

Thanks for the comments. We have added the practical implication for regional development as follows (line 280- line 289):

“Besides the common priorities, the key SDGs also differ at provincial level, especially in the trade-off network. Differentiated policies should be considered based on their own key SDGs to maximize their synergies and mitigate their trade-offs. For example, Beijing and Chongqing needs to reduce the dominated trade-off in their rapid economic development (SDG8)⁵⁵⁻⁵⁶. Xinjiang needs to improve the inequalities (SDG10) on the social dimension⁵⁷. Tibet has poor health condition and needs to mitigate the highly negative impact from good health and well-being (SDG3)⁵¹. Heilongjiang and Jilin face high trade-offs from their traditional industry⁵⁸, the most important is to reduce the negative impact from industry (SDG9), and the consumption and production sectors (SDG12), respectively.”

5. Section: Discussion and Policy Implication: The discussion section interprets findings and discusses policy implications, focusing on policy recommendations. Strengths include clear policy implications, use of real-world examples, recognition of regional differentiation, and explanation of statistical rigour. Areas for improvement include a need for deeper analysis, exploration of cross-boundary interactions, better connection to the broader global context, more specific policy recommendations, and concise concluding remarks.

Issue 19: While the section discusses SDGs' common and differentiated priorities, it could benefit from a deeper analysis of why certain goals are more challenging or have higher trade-offs in specific regions. Exploring the underlying causes and context-specific factors would enhance the discussion.

Response: Revised as suggested.

Thanks for the comments. We have added the deeper analysis of the underlying causes and context in the discussion as follows (line 321- line 344):

“The total carbon emissions in eastern China were significantly higher than those in central and western regions. The main reason was that the eastern region had a developed economy and a high demand for energy. For example, the rapid urbanization and industrial scale-up in the eastern region required a large amount of energy support, making the trade-off in SDG13 (Combating Climate Change) in Eastern China at a high level. The total carbon emissions in northwest China reached 1.61 billion tons in 2019, accounting for 15% of the country's total carbon emissions, with a growth rate of 5.8%³⁰. The reason may be that in order to promote the economic development of the western region, China proposed a series of policies based on the “Western Development” strategy. The implementation of energy policies such as “West-to-east Electricity Transmission” and “West-to-east Gas Transmission” made the northwest region become China's energy important production base⁶¹. At the same time, some industries in the east were encouraged to move to the west, which to a certain extent strengthened the proportion of heavy and chemical industries in the west, which brought an increase in

total carbon emissions in the west, making the trade-off in SDG13 (Combating Climate Change) in northwest China at a high level.

In recent years, the economic development rate of southern China has been significantly higher than that of northern China²⁸, which made the southern region, including East, Central South and Southwest China face more severe gender equality issues (SDG5). During China's planned economy stage (1952-1992), workers' employment allocation and salaries were determined by the Chinese government, which was engineered to result in a small gender wage gap. Yet with a market economy, corporations were given more autonomy, and women faced increased inequality in the workplace⁶². For instance, the gender gap in labor force participation between men and women nearly doubled over the last two decades, rising from 9% in the 1990s to almost 15% in 2020. The increase in gender earnings gap was even more pronounced⁴³.”

Issue 20: The section mentions that SDG synergies and trade-offs may be affected by cross-boundary interactions but does not delve into this aspect in detail. Expanding on how these cross-boundary interactions influence SDG priorities would enrich the discussion.

Response: Revised as suggested.

Thanks for the comments. We have added the discussion on the cross-boundary influence on SDG priorities as follows (line 409- line 420):

“SDG synergies and trade-offs may be affected by cross-boundary interactions through flows of energy, people, technology, financial capital, etc. (Xu et al., 2020). As one example of a spillover effect within China, resource consumption by the more than 21 million residents in Beijing can exacerbate water scarcity (SDG6) and food insecurity (SDG2) in neighbouring Hebei province and even the North China Plain since Beijing mainly relied on resources from its neighbouring regions to support its development. A large part of the water and natural gas used in Beijing was provided through the “South-to-North Water Diversion” and “West-to-East Gas Transmission” Projects. Furthermore, the virtual resources such as water consumed in commodity production, such as food, clothes, etc. were transferred via interregional and international trade (SDG8 & SDG17), which allowed the receiving region to conserve local resources for other needs (Xu et al., 2020). Future research and policy on SDGs interaction networks in China and other countries should account for cross-boundary issues.

Xu, Z., Li, Y., Chau, S. N., Dietz, T., Li, C., Wan, L., ... & Liu, J.. Impacts of international trade on global sustainable development. *Nature Sustainability*, 3(11), 964-971 (2020).

Xu, Z., Chen, X., Liu, J. et al. Impacts of irrigated agriculture on food–energy–water–CO2 nexus across metacoupled systems. *Nat Commun* 11, 5837 (2020). <https://doi.org/10.1038/s41467-020-19520-3>.”

Issue 21: The discussion could be improved by connecting the findings to the broader context of sustainable development and global SDG implementation. How do China's experiences and priorities compare to those of other countries facing similar challenges?

Response: Revised as suggested.

Thanks for the comments. We have added the discussion on connecting our findings to the global context of sustainable development as follows (line 374- line 384):

“Learning from our study, other countries worldwide could also make the common but differentiated SDGs priorities across spatial scales instead of cherry-picking. SDG13 (Climate Action) and SDG5 (Gender Equality) are the key hurdles for China to achieving 2030 agenda. They were also key goals for a successful implementation of SDGs globally^{20,65}. Climate change continues to present a growing and significant global challenge to humanity and the biosphere in the 21st century⁶⁶. Gender inequality is evident all over the world, with serious negative impacts on people's lives⁶⁷. Promoting gender equality and climate actions will accelerate progress across the SDG systems^{29,68}. The world can also learn from the compound positive impacts which China obtained by making progress on SDG1 (No Poverty) and SDG6 (Clean Water and Sanitation), particularly for low-income countries as eliminating poverty and ensuring proper water and sanitation facilities will contribute to accelerate progress across the SDG systems⁶⁸. ”

Issue 22: While the section identifies key challenges and priorities, it could provide more specific policy recommendations or strategies for addressing them. Offering actionable guidance for policymakers would enhance the section's practical utility.

Response: Revised as suggested.

Thanks for the comments. We have added more specific policy recommendations in the discussion as follows (line 345- line 373):

“Large countries such as China can take advantage of fiscal reallocation policies and administrative staffing mechanisms to limit trade-offs and leverage synergies. There was a long history of China using fiscal policy to move resources from more progressive to regions that lagging in socioeconomic development. Through the fiscal transfer payment policy, the central government transferred the fiscal surplus of high-GDP provinces such as Guangdong, Shanghai, and Beijing to Sichuan, Hunan, Hubei, Yunnan and other provinces to ensure education (SDG4), medical care (SDG3), balance the development of public services (SDG1) and narrow the gap on development among regions (SDG10) (Central transfer payments exceeded 10 trillion yuan for the first time, how much money was allocated to each province?, accessed on Oct. 26, 2023). Our results indicate that the health goal in Tibet (SDG3) and the social inequality in Xinjiang (SDG10) has the greatest trade-offs, and are areas where traditional fiscal transfer payments need to be focused. The central government can also consider expanding the service scope of fiscal transfer payments. For example, in the provinces or regions where SDG13 (Combating Climate Change) dominates, fiscal transfer payment support can be considered to increase for Inner Mongolia, Jiangxi, Ningxia, Shaanxi, and Shanxi to address the challenge. In terms of SDG5 (Gender Equality), the increased support can occur in Anhui, Gansu, Guangxi, Guizhou, Hainan, Hebei, Hunan, Sichuan, and Yunnan, which have the most prominent trade-offs in SDG5 (Gender Equality), and are fiscally concerned with financial income less than expenditure. While Fujian, Liaoning, Shandong, Zhejiang, and Jiangsu can increase their support for SDG5 based on their own finances as they all have financial surplus.

Similarly, there is also the potential—more so that in a modern democracy—to move local leaders to a region where there is a need for innovative thinking on the links between climate and gender. Based on the performance highlights and comprehensive qualities of provincial leaders in the past, leaders can be moved to other provinces to promote learning from the experience of advanced provinces in dealing with SDGs in trade-offs and synergies (Li et al., 2022). For example, Tianjin has high synergies in SDG5 (Gender Equality), which is worthy of reference by all other provinces. Tibet with the highest trade-offs in SDG3 (Good Health and Well-being) can learn from the experience of Yunnan, which has greater synergy benefits in the health goal. Tianjin and Hubei having the highest trade-offs in SDG2 (Zero Hunger) can learn from Henan which obtain the highest synergies. Heilongjiang with the highest trade-offs in SDG9 can learn from the experience of Anhui and Qinghai with high synergies in this goal.

Central transfer payments exceeded 10 trillion yuan for the first time, how much money was allocated to each province? <https://baijiahao.baidu.com/s?id=1763142399541846936&wfr=spider&for=pc>, accessed on Oct. 26, 2023 [In Chinese].

Li, Y., Shao, X., Tao, Z., & Yuan, H.. How local leaders matter: Inter-provincial leadership transfers and land transactions in China. *Journal of Comparative Economics*, 50(1), 196-220 (2022).”

Issue 23: The section could benefit from a concise summary or concluding remarks that reiterate the main policy implications and the significance of the research.

Response: Revised as suggested.

Thanks for the comments. We have added the concluding remarks as follows (line 421- line 432):

“With over two decades of data over China, we provide new insights into the common but differentiated SDGs priorities at provincial, regional, and national levels through interaction networks. In total, 19 provinces show the highest trade-offs in SDG13 (Combating Climate Change) or SDG5 (Gender Equality) consistent with the national level, with a difference rate of 12/31 while 24 provinces show the highest synergies in SDG1 (No Poverty) or SDG6 (Clean Water and Sanitation) consistent with the national level, with the difference rate of 7/31. These common but differentiated SDG priorities reflect that in the meantime of ensuring a coordinated national response, China should pay more attention to the actual situation of each province, so that provincial governments can formulate more targeted policies in line with provincial SDGs priorities towards high-quality sustainable development. Our study also provides China's example for determining priorities and improving the balance and integrity of measures towards achieving the SDGs to the other countries in the world.”

Reviewers' Comments:

Reviewer #1:

Remarks to the Author:

The revised version of the paper has clarified the issues/ambiguities that I pointed out in the first review. I appreciate the authors' effort in addressing my queries either by revising the text or, in some instances, conducting additional experiments or assessments. The methodology part is much clearer now. The paper can make a good contribution to the scientific literature on the understanding of the interconnectedness of the SDGs, and their synergies/trade-offs at national and subnational levels, including policy implications (SDG prioritization).

Reviewer #2:

Remarks to the Author:

Thank you to the authors for engaging seriously with the previous comments. I think that the paper and the abstract (and title) has improved considerably over previous versions. I particularly appreciate the efforts to discuss the policy/institutional implications of having a varying synergies/trade-offs within a large country like China with a unique political and economic system.

While I am being a little picky here, I would like to suggest two other small changes.

1) My sense is that the newly added text that offers the context on different provinces is not written as well as the other parts of the paper (see examples below). I would suggest that there is another go to improve the readability of especially text from pp. 12 to 16.

Example of text that could be rewritten.

The total amount of medical and health resources was insufficient and unevenly 248 distributed, and medical service capabilities are weak. From the perspective of institutional 249 mechanisms, problems were still prominent, such as extensive management methods, insufficient 250 strict implementation of the medical system, and insufficient procurement of urgently needed 251 drugs. From the perspective of behavioral concept, the public's awareness of health was not 14 strong, and the problem of diseases caused by unhealthy lifestyles was obvious⁵¹. Through promoting the 268 combination of the market and the government leverages, different programs were implemented, including special poverty alleviation, industry poverty alleviation, social poverty alleviation⁵³ 269 . 270 From 2013 to 2022, a total of 2.502 million poor people in the province were lifted out of poverty, and per capita disposable income increased by more than 2.6 times⁵⁴ 271 . Through targeted 272 poverty alleviation and poverty alleviation efforts, Guangdong was at the forefront of the country, 273 making SDG1 (No Poverty) the highest goal in synergies.

2) This is slightly new suggestion, but might want to consider noting that China has a long history on working on co-control between climate and air pollution that is increasingly focusing on synergies— and this suggests scope for applying the synergies logic more to the local level.

See this article: <https://www.mdpi.com/2225-1154/11/12/234>

Reviewer #3:

Remarks to the Author:

Review Report: Acceptance of Manuscript Titled "Climate Action and Gender Equality in China's Sustainable Development"

The authors have responded effectively to the feedback provided by me, addressing specific points related to the abstract, introduction, results, and the section on the similarity and differences of SDG priorities among provinces. The revisions have significantly strengthened the paper and improved its overall clarity and comprehensiveness.

1. Abstract :

- The authors have successfully incorporated specific findings and quantitative data in the abstract, making it more informative and enticing to potential readers.
- The implications for China's sustainable development are now clearly articulated, providing a more explicit connection between the research findings and their significance.

2. Introduction:

- The introduction has been improved by adding relevance to gender and climate goals in the context of China's sustainable development.
- The transition to the methodology section has been made smoother, providing a clearer link between the research gap and the study's aims.

3. Results:

- A more detailed explanation of metrics, including the ABS(R) and Ratio, has been added, enhancing the clarity of the methodology for readers unfamiliar with these measures.
- The interpretation of results has been expanded, providing deeper insights into why certain SDGs exhibit higher trade-offs or synergies, and the practical implications for China's sustainable development.
- The authors have appropriately addressed the issue of data quality and reliability, acknowledging the challenges and limitations in data collection at both national and sub-national levels.

4. Section on Similarity and Differences of SDG Priority among Provinces:

- The authors have provided a deeper interpretation of results, explaining the reasons behind variations in SDG priorities among provinces and the implications for regional development.
- The connection to research questions has been strengthened, explicitly linking the provincial-level findings to the research gap on SDG interactions in China.

Overall, the revisions have significantly improved the manuscript, addressing the my comments comprehensively. The paper now offers a more robust analysis of the complex interactions among SDGs in China, providing valuable insights for policymakers and researchers. Therefore, I recommend accepting the manuscript for publication in its current form. The authors have successfully incorporated reviewer feedback, resulting in a well-structured and informative contribution to the field of sustainable development.

Response to the Editor and Reviewers

We would like to express our gratitude to the editor and anonymous reviewers for their valuable comments and suggestions for improving the quality of the paper. We have carefully considered all the points raised by them. We are providing detailed point-by-point responses to all questions and recommendations by the editors and reviewers. In the responses below, the red fonts are the revised texts with the column number in the revised version.

Part 1: Response to reviewer 1

The revised version of the paper has clarified the issues/ambiguities that I pointed out in the first review. I appreciate the authors' effort in addressing my queries either by revising the text or, in some instances, conducting additional experiments or assessments. The methodology part is much clearer now. The paper can make a good contribution to the scientific literature on the understanding of the interconnectedness of the SDGs, and their synergies/trade-offs at national and subnational levels, including policy implications (SDG prioritization).

Response: Thank you. We appreciate the reviewer's valuable comments for improving our manuscript.

Part 2: Response to reviewer 2

Thank you to the authors for engaging seriously with the previous comments. I think that the paper and the abstract (and title) has improved considerably over previous versions. I particularly appreciate the efforts to discuss the policy/institutional implications of having a varying synergies/trade-offs within a large country like China with a unique political and economic system.

Response: Thank you. We appreciate the reviewer's valuable comments for improving our manuscript.

While I am being a little picky here, I would like to suggest two other small changes.

1) My sense is that the newly added text that offers the context on different provinces is not written as well as the other parts of the paper (see examples below). I would suggest that there is another go to improve the readability of especially text from pp. 12 to 16.

Example of text that could be rewritten.

The total amount of medical and health resources was insufficient and unevenly 248 distributed, and medical service capabilities are weak. From the perspective of institutional 249 mechanisms, problems were still prominent, such as extensive management methods, insufficient 250 strict implementation of the medical system, and insufficient procurement of urgently needed 251 drugs. From the perspective of behavioral concept, the public's awareness of health was not 14 strong, and the problem of diseases caused by unhealthy lifestyles was obvious⁵¹

Through promoting the 268 combination of the market and the government leverages, different programs were implemented, including special poverty alleviation, industry poverty alleviation, social poverty alleviation⁵³ 269 . 270 From 2013 to 2022, a total of 2.502 million poor people in the province were lifted out of poverty, and per capita disposable income increased by more than 2.6 times⁵⁴ 271 . Through targeted 272 poverty alleviation and poverty alleviation efforts, Guangdong was at the forefront of the country, 273 making SDG1 (No Poverty) the highest goal in synergies.

Response: Thanks for the comments. We have revised the all the text from pp. 12 to 16 to improve the readability. The revised text is in line 161-259 in the revised version and listed below:

“At the provincial level, we found the differences in SDG synergies and trade-offs within China. In total, there were 244-872 pairs in synergies and 62-380 pairs in trade-offs with the averaged ABS(R) of 0.92-0.96 and 0.9-0.94 (Bonferroni corrected $p < 0.05$ and $ABS(R) > 0.6$) (see Supplementary Table 1 and Supplementary Data 4 of “Synergies and trade-offs” for more details on each province). At the national level, we could assess the overall situation across China. Among goals, SDG13 (Climate Action) and SDG5 (Gender Equality) had the lower hub scores in synergies on average (0.19 and 0.34) (Fig. 4(a), 4(c) and 5(a)) and higher in trade-offs (0.76 for both) for 19 provinces consistent with the national level (Fig. 4(b), 4(d) and 5(b)). 14 of these provinces had the highest trade-offs in SDG5 (Gender Equality), and 5 of them had the highest trade-offs in SDG13 (Climate Action) (see supplementary text of the results at the provincial level in SI for more details on the 19 provinces and Supplementary Fig. 2 for the trade-off networks at the provincial level with source data in Supplementary Data 1). The goal with the highest trade-offs differed among the other 12 provinces. They included SDG8 (Decent Work and Economic Growth) for Beijing and Chongqing, SDG10 (Reduced Inequalities) for Xinjiang, SDG3 (Good Health and Well-being) for Tibet, SDG9 (Industry, Innovation, and Infrastructure) for Heilongjiang, SDG12 (Responsible Consumption and Production) for Jilin, SDG17 (Partnerships for the Goals) for Guangdong, Shanghai, and Qinghai, SDG16 (Peace, Justice, and Strong Institutions) for Henan, and SDG2 (Zero Hunger) for Hubei and Tianjin (Fig. 4(b) and 4(d), see supplementary Fig. 2 for the trade-off networks at the provincial level with source data in Supplementary Data 1).

At the regional level, SDG13 (Climate Action) and SDG5 (Gender Equality) showed the highest trade-offs except for Northeast China, where SDG4 (Quality Education) had the most considerable trade-offs (Fig. 6(b), Fig. 7(c) and 7(d)). The southern regions had a higher trade-off in SDG5 (Gender Equality) than the northern regions (Fig. 7(c)). However, SDG13 (Climate Action) had an opposite pattern between north and south in terms of trade-off (Fig. 7(d)).

We found that most of the SDGs had higher synergies than trade-offs. Among them, SDG1 (No Poverty) and SDG6 (Clean Water and Sanitation) showed high scores in synergies (0.98 and 0.97) (Fig. 4(a), 4(c) and 5(a)) and low scores in trade-offs (0.35 and 0.42) for 24 provinces consistent with the national level. Among them, there were the highest synergies in SDG1 (No Poverty) and SDG6 (Clean Water and Sanitation) for 14 and 10 provinces (Fig. 4(b), 4(d) and 5(b), see supplementary text of the results at the provincial level in SI for more details on the 24 provinces and supplementary Fig. 1 for the synergy networks at the provincial level with source data in Supplementary Data 1). The goal with the highest synergies differed among the other 7 provinces. They included SDG2 for Henan, SDG11 (Sustainable Cities and Communities) for Inner Mongolia and Jilin, SDG7 (Affordable and Clean Energy) for Jiangxi, SDG4 (Quality Education) for Sichuan, Tibet, and Xinjiang (Fig. 4(a) and 4(c), see supplementary Fig. 1 for the synergy networks at the provincial level with source data in

Supplementary Data 1). In all six regions, SDG1 (No Poverty) and SDG6 (Clean Water and Sanitation) had the most synergies (Fig. 6(a), Fig. 7(a) and 7(b)). These findings reflected the need for common but different priorities for SDGs in China at the national and provincial levels.

Each province or region had different synergy and trade-off priorities due to its own geographical location, resource endowment, climatic condition, topography, and historical development. The varied SDG priorities reflected that China should pay more attention to the actual situation of each province to ensure a coordinated national response. In doing so, provincial governments could formulate more targeted policies aligning with regional and national SDG priorities towards accelerating sustainable development.

Here we took Tibet, which had the lowest GDP in China, as an example to discuss the provincial-level trade-offs. Restricted by the natural, geographical, climatic, and historical factors, Tibet's overall medical and health services development was still lagging. The total medical and health resources were insufficient and unevenly distributed, and the medical service capabilities were weak. Problems still existed in institutional mechanisms, such as extensive management manner, insufficient strict implementation of the medical system, and insufficient procurement of much-needed drugs. From the perspective of behavioral concept, the public's awareness of health was not strong, and the problem of diseases caused by unhealthy lifestyles was obvious⁵¹. Looking at the disease spectrum, the incidence rates of AIDS, tuberculosis, and hepatitis B per 100,000 people increased from 0.02, 75.07, and 52.62 in 2002 to 1.34, 150.13, and 107.29 in 2020, respectively⁵². The combined constraints of the natural, geographical, climatic, and historical factors made SDG3 (Good Health and Well-being) in Tibet have the highest trade-offs.

We took Guangdong as an example to discuss the provincial level synergies. Guangdong, at the forefront of reform and opening up, had the highest GDP in China. However, there was a large amount of poverty in the mountainous areas in the north and the underdeveloped areas in the west due to remote geographical location, numerous mountainous areas, insufficient transportation infrastructure, and single industrial structure. To this end, the Guangdong government integrated the targeted poverty alleviation campaign into the overall economic and social development plans for the overall planning, fully leveraging the synergies between SDG1 (No Poverty) and different SDGs. The government established a poverty reduction governance pattern emphasizing mutual promotion and focused on stimulating the endogenous motivation to eliminate poverty. Through precise strategies, policies were implemented accurately and targeted to villages, households, and people. Different programs were implemented by promoting the combination of the market and the government leverages, including special poverty alleviation, industry poverty alleviation, and social poverty alleviation⁵³. From 2013 to 2022, 2.5 million poor people in the province were lifted out of poverty, and the disposable income per capita increased by more than 2.6 times⁵⁴. Through targeted poverty alleviation and poverty alleviation efforts, Guangdong was at the forefront of the country, making SDG1 (No Poverty) have the highest synergies. In the future, Guangdong needs to continue to leverage the synergistic advantages of SDG1 (No Poverty). Doing so will gradually narrow the income gap by further getting rid of relative poverty and realizing the rural revitalization.

Our results highlighted the need to prioritize different SDGs among Chinese provinces and regions based on understanding SDG interactions at various spatial levels. There were some common priorities, but the key SDGs differed at the provincial level, especially in the trade-off networks. Differentiated policies should be considered based on SDG interactions at the provincial and regional levels to

maximize synergies and mitigate trade-offs. For example, Beijing and Chongqing need to reduce the dominant trade-offs generated by their rapid economic development (SDG8)⁵⁵⁻⁵⁶. Xinjiang need to improve the inequalities (SDG10) at the social dimension⁵⁷. Tibet had poor health conditions and need to mitigate the highly negative impact of good health and well-being (SDG3)⁵¹. Heilongjiang and Jilin faced high trade-offs from their traditional industries, the most important was to reduce the negative impact from industry (SDG9) and the consumption and production sectors (SDG12)⁵⁸.

At the provincial level, we collected data between 98 and 102 indicators for 30 provinces. For Tibet, with its remote location and difficult conditions, 94 indicators were collected to their maximum extent. After massive efforts on data collections, overall we compiled a set of 102 SDG indicators (118 original indicators and 102 after calculations), 81 targets, and 17 goals, at the provincial level on a yearly basis. This was greater than the number of indicators in the 2022 SDG Index and Dashboards Report (it used 88 indicators to assess China's SDG performances at the national level) and one previous study (it used 88 indicators, 71 targets, and 16 goals to calculate the score of 16 goals in China and analyze the SDGs interactions between 3 general categories based on the goal score)⁵⁹⁻⁶⁰. Our understanding of the SDGs interaction networks in China at provincial, regional, and national levels will evolve as more data becomes available.

”

2) This is slightly new suggestion, but might want to consider noting that China has a long history on working on co-control between climate and air pollution that is increasingly focusing on synergies—and this suggests scope for applying the synergies logic more to the local level.

See this article: <https://www.mdpi.com/2225-1154/11/12/234>

Response: Thanks for the comments. We have added this point in the discussion with the suggested reference and one more. The revised text is in line 332-336 in the revised version and listed below:

“Besides, China has over three-decade long experiences on working on co-control between climate and air pollution that is increasingly focusing on synergies. This suggests the scope for applying the synergies logic more to the local level in China. Other countries can also learn from China’s experiences on the synergic effect between climate and air pollution for their sustainable transformation⁶⁹⁻⁷⁰.”

Cai, Q., Zusman, E., & Meng, G.. The Shift to Synergies in China’s Climate Planning: Aligning Goals with Policies and Institutions. *Climate* **11(12)**, 234 (2023).

Lu, X. et al. Progress of air pollution control in China and its challenges and opportunities in the ecological civilization era. *Eng. J.* 6(12), 1423-1431 (2020).

Part 3: Response to reviewer 3

Review Report: Acceptance of Manuscript Titled "Climate Action and Gender Equality in China's Sustainable Development"

The authors have responded effectively to the feedback provided by me, addressing specific points related to the abstract, introduction, results, and the section on the similarity and differences of SDG priorities among provinces. The revisions have significantly strengthened the paper and improved its overall clarity and comprehensiveness.

1. Abstract :

- The authors have successfully incorporated specific findings and quantitative data in the abstract, making it more informative and enticing to potential readers.
- The implications for China's sustainable development are now clearly articulated, providing a more explicit connection between the research findings and their significance.

2. Introduction:

- The introduction has been improved by adding relevance to gender and climate goals in the context of China's sustainable development.
- The transition to the methodology section has been made smoother, providing a clearer link between the research gap and the study's aims.

3. Results:

- A more detailed explanation of metrics, including the ABS(R) and Ratio, has been added, enhancing the clarity of the methodology for readers unfamiliar with these measures.
- The interpretation of results has been expanded, providing deeper insights into why certain SDGs exhibit higher trade-offs or synergies, and the practical implications for China's sustainable development.
- The authors have appropriately addressed the issue of data quality and reliability, acknowledging the challenges and limitations in data collection at both national and sub-national levels.

4. Section on Similarity and Differences of SDG Priority among Provinces:

- The authors have provided a deeper interpretation of results, explaining the reasons behind variations in SDG priorities among provinces and the implications for regional development.
- The connection to research questions has been strengthened, explicitly linking the provincial-level findings to the research gap on SDG interactions in China.

Overall, the revisions have significantly improved the manuscript, addressing the my comments comprehensively. The paper now offers a more robust analysis of the complex interactions among SDGs in China, providing valuable insights for policymakers and researchers. Therefore, I recommend accepting the manuscript for publication in its current form. The authors have successfully incorporated reviewer feedback, resulting in a well-structured and informative contribution to the field of sustainable development.

Response: Thank you. We appreciate the reviewer's valuable comments for improving our manuscript.